# Neurons Speak in Ranges: Breaking Free from Discrete Neuronal Attribution

**Muhammad Umair Haider**                                           *muhammadumairhaider@uky.edu*
*Department of Computer Science*
*University of Kentucky*
*Kentucky, USA*

**Hammad Rizwan**                                                   *hammad.rizwan@dal.ca*
*Department of Computer Science*
*Dalhousie University*
*Halifax, Canada*

**Hassan Sajjad**
*Department of Computer Science*
*Dalhousie University*
*Halifax, Canada*

**Peizhong Ju**
*Department of Computer Science*
*University of Kentucky*
*Kentucky, USA*

**A.B. Siddique**
*Department of Computer Science*
*University of Kentucky*

**Reviewed on OpenReview:** *https://openreview.net/forum?id=AukyIhfBuW*

## Abstract

Pervasive polysemanticity in large language models (LLMs) undermines discrete neuron–concept attribution, posing a significant challenge for model interpretation and control. We systematically analyze both encoder and decoder-based LLMs across diverse datasets, and observe that even highly salient neurons for specific semantic concepts consistently exhibit *polysemantic* behavior. Importantly, we uncover a consistent pattern: concept-conditioned activation magnitudes of neurons form distinct, often Gaussian-like distributions with minimal overlap. Building on this observation, we hypothesize that interpreting and intervening on concept-specific activation ranges can enable more precise interpretability and targeted manipulation in LLMs. To this end, we introduce NeuronLens[1], a novel range-based interpretation and manipulation framework that localizes concept attribution to activation ranges within a neuron. Extensive empirical evaluations show that *range-based interventions* enable effective manipulation of target concepts while causing substantially less collateral degradation to auxiliary concepts and overall model performance compared to neuron-level masking.

## 1 Introduction

Neuron interpretation aims to uncover how individual neurons encode semantic concepts and contribute to model outputs. Recent work has made significant progress in this direction by identifying neurons that are

---

[1]Code can be found at https://github.com/MuhammadUmairHaider/NeuronLens.

strongly associated with specific concepts or model behavior (Dalvi et al., 2019a; Antverg & Belinkov, 2022; Conmy et al., 2023; Marks et al., 2024). Common approaches include maximal activation analysis (Foote et al., 2023; Frankle & Carbin, 2019), which links neurons to inputs that produce the highest activations, probe-based methods (Dalvi et al., 2019a;b) that employ auxiliary classifiers to assess neuron–concept associations, and probeless approaches (Antverg & Belinkov, 2022) that infer such associations directly from neuron activations.

These methods typically rely, explicitly or implicitly, on discrete neuron-to-concept mappings, assuming that entire neurons encode single concepts. However, neurons frequently exhibit polysemanticity; the ability to encode multiple, seemingly unrelated concepts (Lecomte et al., 2024; Marshall & Kirchner, 2024). Given this heterogeneous encoding of concepts, traditional approaches can lead to unintended consequences when manipulating neurons, as changes intended for one concept may inadvertently affect others encoded by the same neuron, and suboptimal interpretations of concepts (Sajjad et al., 2022).

To better understand how polysemanticity manifests at the neuron level, we conduct a systematic analysis of neuron activations in both encoder- and decoder-based LLMs across multiple datasets. Focusing on neurons identified as salient for specific concepts, we observe that these neurons consistently exhibit polysemantic behavior, often responding to multiple concepts. Through qualitative and quantitative analysis, we further discover that concept-conditioned neuronal activation magnitudes form *Gaussian-like distributions*, with minimal overlap across different concepts. This observation suggests that although individual neurons are polysemantic, the activations associated with a given concept tend to concentrate within a specific band of activation magnitudes. That is, each concept is typically expressed within a characteristic activation range of a neuron, even when the same neuron participates in encoding multiple concepts.

Building on this observation, we introduce NeuronLens, a range-based attribution framework for neuron interpretation and manipulation, illustrated in Figure 1. Rather than attributing an entire neuron to a single concept, NeuronLens identifies and maps specific activation ranges within a neuron's distribution to individual concepts. Specifically, for a given concept, NeuronLens calculates an activation range that captures the majority of concept-conditioned activations, enabling precise attribution, which in turn allows for interventions that operate selectively within this range while leaving other activation regimes of the neuron unaffected. To causally validate our approach, we run extensive concept-erasure experiments. Across settings, our method reduces unintended interference on auxiliary concepts by up to 25 percentage points (-14 on Llama) than full neuron masking. Additionally, our range-based interventions preserve general capabilities of models, maintaining or improving performance on MMLU benchmark in most settings, while incurring only minor increases in language modeling perplexity on Wikipedia text.

Our work makes the following contributions. (1) We show that neuronal activations in LLMs form concept-conditioned, Gaussian-like distributions, with often exhibiting limited overlap across concepts. (2) We empirically demonstrate that concept-specific activation ranges are consistently identifiable within polysemantic neurons, providing a more fine-grained handle for neuron-level interpretation than discrete neuron-to-concept attribution. (3) We introduce NeuronLens to interpret and causally validate concept-specific activation ranges for targeted interventions, enabling precise manipulation of target concepts while reducing unintended interference compared to neuron-level masking.

## 2 Neuron Interpretation Analysis

### 2.1 Preliminaries

**Neuron.** We refer to the output of an activation as a neuron. In a transformer model, we consider neurons of hidden state vectors of different transformer layers. Formally, given a hidden state vector $\mathbf{h}^l \in \mathbb{R}^d$ of size $d$ produced by layer $l$, $h_j^l$ denotes its $j$-th neuron, i.e., the $j$-th component of $\mathbf{h}^l$.

**Concept.** A concept $c \in C$ is a high-level semantic category that groups each input instance (or components of every instance), where $C$ is the set of all concepts. For example, in a language task, a sentence can be categorized into four types: declarative, interrogative, imperative, and exclamatory, where each type is a concept. Words in a sentence can also convey concepts such as nouns, verbs, adjectives, adverbs, and more. We study settings where all inputs are labeled with concepts.

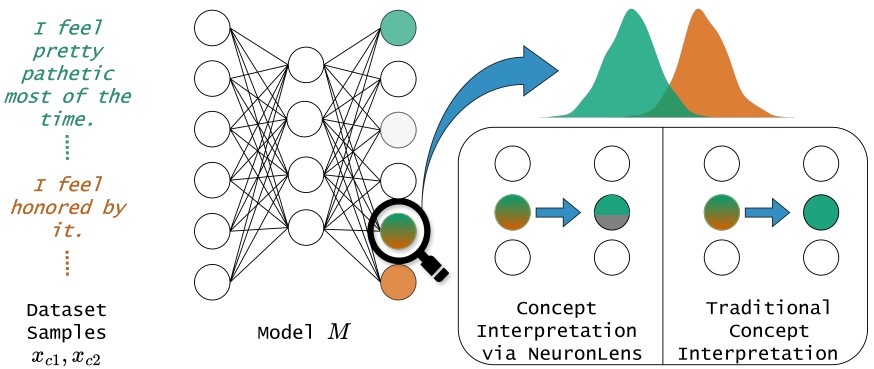

Figure 1: Different concepts induce distinct, Gaussian-like activation distributions within the same neuron, enabling range-based concept attribution and intervention.

**Saliency Ranking.** A saliency ranking orders the importance of neurons based on some saliency metric. For a hidden state vector $\mathbf{h}^l$, we denote the value of the saliency metric for the $j$-th neuron with respect to a concept $c$ as $s_j^c$. The saliency ranking $(r_c(1), r_c(2), \cdots, r_c(d))$ is a permutation of the indices of neurons $(1, 2, \cdots, d)$, where $r_c(j) < r_c(i)$ if $s_j^c > s_i^c$. The saliency metric is usually predefined, such as absolute neuron activation values.

**Concept Learning.** Given a hidden state vector $\mathbf{h}^l$ as input, the associated concept can be the output of an appended neural network (e.g., several fully connected layers). The parameters of this appended neural network can be trained using training samples labeled with concepts.

## 2.2 Datasets and Models

**Datasets.** For various experiments in this work, we utilize the following datasets: sentiment analysis (IMDB (Maas et al., 2011)), (SST2 (Socher et al., 2013)), emotion detection (Dair-Ai/Emotions (Saravia et al., 2018)), news classification (AG-News (Zhang et al., 2015)), and article content categorization (DBPedia-14 (Zhang et al., 2015)). Moreover, we use the MMLU benchmark (Hendrycks et al., 2021) to evaluate the general capabilities of LLMs and Wikipedia texts (Foundation) for open-ended generation. Details of concepts tested, along with examples, are provided in Appendix O Table 33, and Table 34.

**Models.** This work employs both encoder and decoder-based models, including pretrained Llama-3.2-3B (Grattafiori, 2024), fine-tuned BERT (Devlin et al., 2019), DistilBERT (Sanh et al., 2020), and GPT-2 (Radford et al., 2019).

## 2.3 Salient Neurons Extraction

We record activations for training samples of different concepts to perform neuron interpretation. Specifically, if we want to interpret neurons of $\mathbf{h}^l$ (hidden vector at layer $l$), we perform a forward pass using the training dataset and store the values of $\mathbf{h}^l$ and the associated concepts of all samples into a set $H^l$. The set $H^l$ is further partitioned into $H_c^l$ for all concepts $c \in C$. Such preparation is common in the relevant literature (Dalvi et al., 2019c;b; Antverg & Belinkov, 2022).

We explore standard neuron saliency approaches, namely max activations (Frankle & Carbin, 2019), probless (Antverg & Belinkov, 2022), and probe analysis (Dalvi et al., 2019b). Details of these approaches are provided in Appendix C. To evaluate these saliency methods, we employ a concept erasure task that masks the neurons identified as salient by each method and measures their ability to suppress a target concept while minimally affecting auxiliary concepts. Table 6 in Appendix C provides the results for this experiment. Notably, all evaluated saliency methods exhibit degradation in auxiliary concepts when salient neurons for a target concept are masked. One explanation for such deterioration in auxiliary concepts is due to the

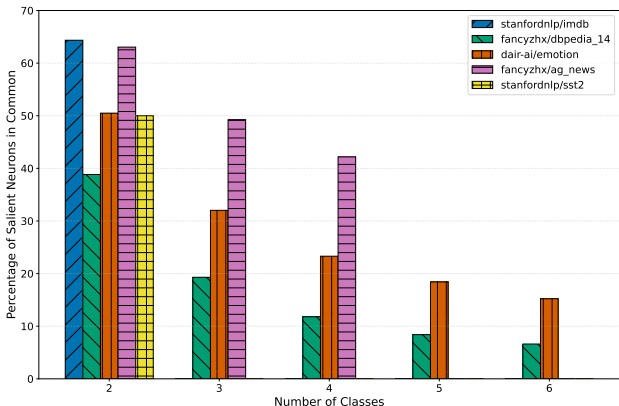

Figure 2: Overlap in top 30% salient neurons in GPT-2 model across classes in various datasets.

*polysemantic* neurons. That is, if individual neurons encode multiple concepts, interventions targeting a specific concept can inadvertently disrupt other concepts co-represented within the same neurons.

While any saliency method can be used to identify influential neurons, *we adopt max activation ranking throughout this work because it provides the strongest targeted suppression* as compared to other evaluated methods while exhibiting comparable degradation in auxiliary concepts.

## 3 Polysemanticity

Polysemanticity often arises when models must represent more features than their capacity allows or due to specific training paradigms (Anthropic, 2023). Training methods like subword tokenization, designed to reduce vocabulary size and model complexity, lead to context-dependent token splits, causing activations to encode multiple meanings (Sennrich et al., 2016; Elhage et al., 2022; Meng et al., 2022). Additionally, Lecomte et al. (2024) show that even with sufficient capacity, weight initializations can induce polysemanticity by placing neurons near multiple conceptual regions. Irrespective of its cause, the polysemanticity of neurons, including salient neurons that encode multiple concepts, challenges the discrete neuron-to-concept attribution paradigm, which maps a concept to an entire neuron.

**Polysemanticity in Salient Neurons.** To study polysemanticity in salient neurons, we consider a neuron to be polysemantic if it appears salient for more than one concept. We calculate the overlap between the top 30% salient neurons (i.e., max activations) in a model across different classes of diverse datasets. Figure 2 presents the results on GPT-2 model using five different datasets. We observe a considerable overlap in salient neurons between concepts (i.e., classes). For instance, in the case of a two-class dataset, IMDB, the overlap in salient neurons is more than 60%, showing a high degree of polysemanticity.

## 4 Neuronal Activation Patterns

The prevalence of polysemanticity in salient neurons raises a natural follow-up question: *how do multiple concepts manifest within the activations of a single neuron?* To investigate this question, we analyze the activation patterns of salient neurons extracted via maximal activation, including those that exhibit polysemantic behavior. Specifically, we examine neuron activations conditioned on individual concepts across multiple datasets. Similar to Gurnee et al. (2024), we observe that neuronal activations form Gaussian-like distributions. Importantly, our findings further indicate that for a given salient neuron, *activations associated with different concepts form distinct Gaussian-like distributions with limited overlap*, suggesting that concepts tend to occupy characteristic regions within a neuron's activation spectrum. In the following, we present qualitative and quantitative analyses of the concept-conditioned activation patterns.

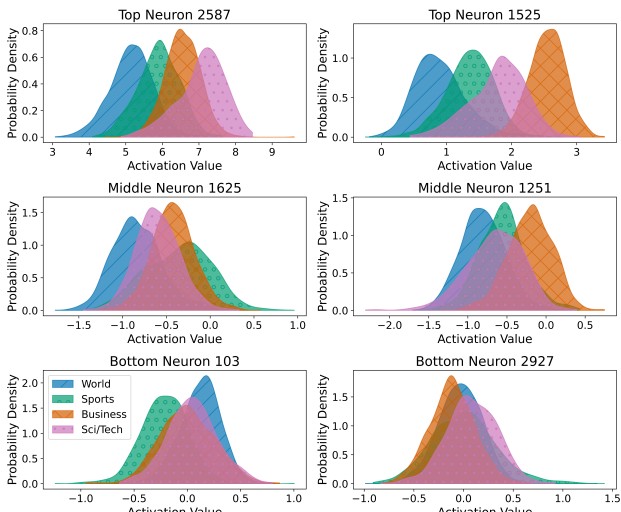

Figure 3: Neuronal activation patterns of six neurons in the Llama model on *AG-News*. Top: neurons sampled from the highest-activating group; middle: from the mid-ranked; bottom: from the lowest-activating group.

Table 1: Skewness, kurtosis, and Kolmogorov-Smirnov test results across various datasets for Llama model.

| Dataset | Skewness | Kurtosis | KS-Test |
|---|---|---|---|
| stanfordnlp/imdb | 0.0003 | 3.9251 | 0.9995 |
| fancyzhx/dbpedia_14 | 0.0068 | 3.9916 | 0.9133 |
| dair-ai/emotion | 0.0000 | 3.1609 | 0.9988 |
| fancyzhx/ag_news | -0.0020 | 3.4079 | 0.9995 |
| stanfordnlp/sst2 | -0.0034 | 3.5890 | 0.9993 |

## 4.1 Qualitative Analysis

To visually demonstrate that neuron activations for a concept $c$ follow a Gaussian-like distribution, we extract model activations as described in § 2. Using saliency ranking $r_c$ for the dataset, we examine neurons from different ranking positions in the Llama model on the AG-News dataset. In Figures 3, we visualize these distributions using Kernel Density Estimation (KDE).

It reveals that the activations are Gaussian-like for different concepts, where salient neurons demonstrate distinct activation patterns with limited overlap, middle-ranked neurons show a higher degree of overlap than the top ones, whereas non-salient neurons (bottom two) exhibit the highest overlap in activation distributions. Additional analysis for different types of polysemantic neurons for the GPT-2 model is provided in Appendix D. For example, figure 6 in Appendix D visualizes two distinct types of polysemantic neurons. One maintains partially separable activation patterns despite being polysemantic; the other exhibits completely overlapping activations. Figure 7 in Appendix D, we present a broader analysis of 7 neurons from the polysemantic subset for 14 total classes. We observe that most polysemantic neurons exhibit limited overlap between concepts at the distributional level with distinct means.

**Text Analysis.** We complement the activation analysis with a qualitative text analysis that examines representative samples associated with different regions of concept-conditioned activation distributions. Specifically, for selected neurons showcased in Figure 3, we analyze samples drawn from the lower boundary, mean, and upper boundary of their activation distributions. Table 2 presents representative examples for neurons 1525 and 2587, which are predominantly associated with *Sci/Tech* and *Business* concepts, respectively. For neuron 1525, samples near the mean of the distribution focus on core *Sci/Tech* terminology, while samples near the upper boundary reflect a mix of *Sci/Tech* and *Business* concepts (e.g., "subscription plans"). In contrast, samples near the lower boundary tend to blend *Sci/Tech*, *Sports* (e.g., "marathon"), and *World* themes. A similar pattern is observed for neuron 2587: samples near the mean focus on core *Business* topics,

Table 2: Semantic concepts are highlighted by class color: **World (Blue)**, **Sports (Green)**, **Business (Orange)**, and **Sci/Tech (Purple)**. For each neuron, examples are shown from the lower boundary, mean, and upper boundary of the neuron activations, illustrating how semantic content varies across activation magnitudes. The table shows shortened texts; full reference texts are provided in Appendix table 8.

| Neuron | Point | Boundary Affinity | Representative Examples (Concepts Highlighted) |
|---|---|---|---|
| 1525 | Lower Boundary | World / Sports | "**Gene Tweaking** Turns... **Racers**...**single gene** turned...**marathon racers** that could **run for hours**." 
 "**Insecure elections**...**marching**...**soldiers overseas**...ballots via **e-mail**...**PDF**...**Defense Department**." 
 "**IBM**'s **supercomputer** breaks **world's fastest**...**NEC**'s Earth Simulator...based at **Yokohama, Japan**." |
| | Mean | Sci/Tech | "...**powerful chip**...**microprocessor**...appearing in **video game consoles** and **high-definition T-Vs**." 
 "**AMD**...**Dual-Core Opteron**...faster than **single-core chips**...fit in existing **server designs**." 
 "**Arm reveals multimedia**...**Microprocessor** designer...**multimedia technology** for **mobile electronics**." |
| | Upper Boundary | Business | "iPass..**Flat-Rate Pricing Plans** for US **Wi-Fi Hotspot**...**subscription plans** for use..**Wi-Fi connectivity**." 
 "NTT DoCoMo...**Tokyo Stock Exchange**...**procuring mobile phone handsets** made by **Motorola Inc**." 
 "Omnipod...**companies of all sizes**...enhancements...**Web-based client**...a **persistent-chat feature**." |
| 2587 | Lower Boundary | World / Sports | "**Put Me in, Coach!**. Coach joins the **S&P 500**...leather in the weather as the **global stock market** reacts." 
 "The **London Stock Exchange** plans...**Asia headquarters** in **Hong Kong**...eyeing **stock listings abroad**." 
 "**Halliburton** closes higher... **Army**'s decision...**shares closed higher**...**contract** to supply **US troops in Iraq**." |
| | Mean | Business | "UBS...**capital markets unit** for **$265 million in cash**, strengthening...**brokerage business**" 
 "**Applied Materials**... **shares** were off... in **trading**... wavered around **break-even**... **results** were announced." 
 "Brown-Forman...**jump in earnings** as **aggressive marketing** boosted **sales of premium spirits**." |
| | Upper Boundary | Sci/Tech | "Arm to pay...**$910m in cash and shares** for...**transistor-level designer for systems-on-a-chip**" 
 "**Oracle sales rise** on **database demand**... focus... obtaining **CRM and ERP software**..." 
 "The...**information technology**...**productivity gains**, **business software, and telecommunications** ..." |

transition toward *World*-related content near the lower boundary (e.g., "Asia headquarters"), and shift toward *Sci/Tech* concepts near the upper boundary (e.g., "database demand"). These examples indicate that *changes in activation magnitude correspond to semantic transitions*, providing qualitative evidence that different concepts occupy characteristic regions within a neuron's activation spectrum. Additional details of the text analysis are provided in Appendix E.

## 4.2 Quantitative Analysis

For each neuron, we collect activation values over samples associated with concept $c \in C$ and compute summary statistics that capture deviations from normality. Specifically, we measure skewness and kurtosis (Joanes & Gill, 1998), and assess goodness-of-fit to a normal distribution using the Kolmogorov–Smirnov (KS) test (Massey Jr, 1951). Table 1 presents the results across all neurons in Llama model. The average skewness is close to 0 across all datasets, indicating strong symmetry (ideal normal distribution: 0), and the average kurtosis is close to 3, nearly identical to the expected value for a normal distribution (3.0).

To assess normality, while accounting for practical significance, we employ the KS test with an effect size threshold of 10%. This approach tests whether the distribution remains within a reasonable bound of normality, rather than testing for perfect normality, which is overly strict for real-world data. For each

neuron, we normalize the activations to zero mean and unit variance, then compute the KS statistic against a standard normal distribution. The KS statistic represents the maximum absolute difference between the empirical and theoretical cumulative distribution functions. Using a threshold of 0.1 (allowing a maximum 10% deviation from normal), we find that close to 100% of the neurons exhibit practically normal distributions. The combination of near-ideal skewness and kurtosis values, visual confirmation through KDEs, and our effect size-based KS tests provides strong evidence that the concept-conditioned activations follow approximately normal distributions.

Additionally, we report layer-wise quantitative statistics for the GPT-2 model in Appendix F.1. As layer depth increases, kurtosis steadily converges toward the Gaussian reference value of 3.0, skewness remains near zero, and the 10% practical-normality score stays close to 1 across the network. A qualitative, layer-wise analysis in Appendix F.2 shows that while concept-conditioned Gaussian-like activation patterns are present across all layers, early layers exhibit substantial overlap between concepts.

Beginning as early as layers 5–6, distinct concept-specific Gaussians emerge and become progressively more separable in later layers, transitioning toward class-specific semantics.

Additional statistics for pairwise AUROC are provided in Appendix D.1 Table 5.

## 5 NeuronLens

Given that neuronal activations exhibit approximately Gaussian-like, concept-conditioned distributions with distinct means, we can interpret and intervene on neurons more precisely by operating on activation ranges rather than attributing entire neurons to individual concepts. This perspective applies regardless of whether a neuron is monosemantic or polysemantic, as it leverages a neuron's activation spectrum. To operationalize this idea, we propose NeuronLens that enables range-based attribution and targeted intervention on neurons.

### 5.1 Range-based Attribution

NeuronLens computes attribution ranges with respect to a concept using means and standard deviations of the neuronal activations. Specifically, given $H_c^l$ that denotes the set of hidden state vector $\mathbf{h}^l(x_c)$ at layer $l$ for all training samples $x_c$ associated with concept $c \in C$, it can calculate the empirical average $\mu \in \mathbb{R}$ and standard deviation $\sigma \geq 0$ of the activation values of the $j$-th neuron $h_j^l$ for all samples associated with the concept $c$ as:

$$\mu = \frac{1}{|H_c^l|} \sum_{\mathbf{h}^l \in H_c^l} h_j^l, \sigma = \sqrt{\frac{1}{|H_c^l|} \sum_{\mathbf{h}^l \in H_c^l} (h_j^l - \mu)^2}.$$

Then, it can attribute the activation range of the $j$-th neuron at layer $l$ to the concept $c$ as:

$$\mathrm{AR}(l, j, c) = [\mu - \tau \times \sigma, \ \mu + \tau \times \sigma],$$

where $\tau > 0$ is a hyperparameter which controls the attribution scope. Intuitively, a smaller $\tau$ yields a narrower, conservative range, while a larger $\tau$ expands coverage at the risk of including less concept-specific activations. $AR$ is defined as attribution range associated with a given concept $c$.

### 5.2 Causal Validation

Causal validation is among the most faithful ways to assess attribution correctness (Feder et al., 2021; Vig et al., 2020), as it directly tests whether intervening on the attributed component produces the intended behavioral change, while measuring unintended side effects. Following prior intervention-based validation (Dalvi et al., 2019b; Dai et al., 2022; Dalvi et al., 2019c; Morcos et al., 2018), we utilize concept erasure as a diagnostic intervention to determine the causal effect of identified ranges within neurons for a given concept. The core idea is that if an attribution is salient to a concept, eliminating it should result in the degradation of model's performance on that concept while causing minimal disruption to other concepts. Formally, given a concept-learning model $M$ that maps any input instance $x$ (or part of an instance) to a

concept $M(x) = c \in C$, an ideal intervened model $M'_{\text{ideal}}$ after erasing a target concept $c \in C$ should satisfy the following property:

$$M'_{\text{ideal}}(x) = \begin{cases} \neq M(x) & \text{if } M(x) = c, \\ = M(x) & \text{if } M(x) \neq c. \end{cases} \tag{1}$$

NeuronLens enables precise manipulation of target concepts via identified activation ranges in contrast to the manipulation and ablation of complete neuron activations. Specifically:

$$h_j^l(x) = \begin{cases} \phi(x) & \text{if } h_j^l(x) \in \text{AR}(l, j, c) \\ h_j^l(x) & \text{otherwise} \end{cases} \tag{2}$$

where $\phi(.)$ is a user-specified range-gated activation operator applied only when $h_j^l(x) \in \text{AR}(l, j, c)$. In general, $\phi$ can be any function, instantiated for causal intervention (e.g., clamping, zeroing, scaling) or for explanation/attribution (e.g., masking or tagging activations). For concept erasure, the gated function $\phi(.)$ zeroes out which is in line with Dai et al. (2022); Antverg & Belinkov (2022). Some studies have argued that zeroing out neurons is an overly aggressive intervention that can lead to catastrophic degradation in model performance. In Appendix G, we compare alternative activation interventions, including the *dampening* method (Suau et al., 2024) and *mean replacement* (Suau et al., 2021). Erasure experiments in the main text ablating a specific range of neuronal activation are referred as range-based masking, whereas ablating the complete neurons is referred as neuron masking.

## 5.3 Experimental Setup

We incorporate our intervention at the penultimate layer. Ablation for layer selection is provided in the Appendix F. The training details for the fine-tuned models are provided in Appendix H.

**Metrics.** We causally validate the attribution using two metrics: prediction accuracy and the model's predictive probability as a proxy for confidence score. First, baseline measurements of both accuracy and confidence for all concepts $C$ without any intervention (unmodified model) are established. Post-intervention measurements are recorded for the target concept $c$ and auxiliary concepts (other concepts in the dataset) $c'$. The effectiveness and precision of attribution are assessed through two key metrics: (1) the performance degradation for concept $c$, and (2) the extent of unintended impact on auxiliary concepts $c'$. Throughout our analysis, we denote the accuracy and confidence metrics for concept $c$ as **Acc** and **Conf** respectively, while corresponding measurements for auxiliary concepts $c'$ are represented as **CAcc** and **CConf**. For evaluating the effect of the interventions on LLMs latent capabilities, we utilize **perplexity (PPL)** and zero-shot accuracy on **MMLU**.

**Hyperparameter.** The value of $\tau$ is set to 2.5, assuming a fully Gaussian distribution. This threshold corresponds to a coverage of approximately 98.76% of the distribution, providing a slightly conservative bound for range-based interventions. Ablations for varying $\tau$ are presented in Appendix I. The results indicate that targeted concept deteriorates up to 2.4-2.7 $\tau$ then plateaus, while auxiliary concepts begin to degrade further.

## 5.4 Results and Analysis

Table 3 presents results for the concept erasure task across five benchmark datasets (Class-wise detailed results are provided in Appendix J), demonstrating the effectiveness of range-based masking compared to traditional neuron masking. Results in Table 3 show that multi-class classification tasks with fine-grained labels, such as AG-News, Emotions, and DBPedia-14, exhibit more pronounced effects under intervention. Range-based masking results in significant degradation of primary task performance while preserving auxiliary concept accuracy. This is particularly evident in results for AG-News. On binary classification tasks (IMDB, SST2) provided in Appendix J (Table 25 and 24), both masking approaches show moderate performance drops in targeted concepts. This suggests higher redundancy for coarser binary concepts.

Table 3: Evaluation of selected models on AG-News, Emotions, and DBPedia-14 datasets using activation range and neuron masking techniques. Performance metrics are calculated using class level 10% trimmed means at the class level. Metrics are detailed in § 5.3. For *GPT-2* and *BERT* 50% and for *Llama-3.2-3B* 30% neurons are selected.

| Model | Dataset | Base Values | | | | Neuron Masking | | | | Activation Range Masking (ARM) | | | |
|---|---|---|---|---|---|---|---|---|---|---|---|---|---|
| | | Acc | Conf | CAcc | CConf | $\Delta$Acc | $\Delta$Conf | $\Delta$CAcc | $\Delta$CConf | $\Delta$Acc | $\Delta$Conf | $\Delta$CAcc | $\Delta$CConf |
| BERT | AG-News | 0.948 | 0.929 | 0.948 | 0.929 | -0.271 | -0.590 | 0.012 | -0.074 | -0.261 | -0.590 | **0.013** | **-0.009** |
| | Emotions | 0.894 | 0.834 | 0.917 | 0.876 | -0.291 | -0.633 | 0.013 | -0.265 | -0.279 | -0.635 | **0.014** | **-0.069** |
| | DBPedia-14 | 0.992 | 0.991 | 0.990 | 0.989 | -0.028 | -0.786 | 0.000 | -0.017 | -0.015 | -0.766 | **0.000** | **-0.000** |
| GPT-2 | AG-News | 0.945 | 0.933 | 0.945 | 0.933 | -0.871 | -0.877 | -0.155 | **-0.163** | -0.849 | -0.862 | **-0.063** | -0.223 |
| | Emotions | 0.905 | 0.892 | 0.930 | 0.919 | -0.735 | -0.738 | -0.103 | -0.103 | -0.737 | -0.739 | **-0.044** | **-0.046** |
| | DBPedia-14 | 0.993 | 0.990 | 0.990 | 0.988 | -0.810 | -0.845 | -0.154 | -0.177 | -0.782 | -0.825 | **-0.015** | **-0.031** |
| Llama | AG-News | 1.000 | 0.744 | 1.000 | 0.744 | -0.934 | -0.725 | -0.660 | -0.572 | -0.935 | -0.725 | **-0.484** | **-0.454** |
| | Emotions | 0.815 | 0.472 | 0.823 | 0.477 | -0.795 | -0.470 | -0.696 | -0.429 | -0.797 | -0.469 | **-0.594** | **-0.404** |
| | DBPedia-14 | 1.000 | 0.533 | 1.000 | 0.563 | -0.992 | -0.528 | -0.912 | -0.445 | -0.986 | -0.527 | **-0.663** | **-0.354** |

Table 4: Evaluation of latent capabilities of Llama model after applying neuron and range-based masking. Base PPL = 7.007; Base MMLU = 0.53.

| Dataset | Neuron Masking | | ARM | |
|---|---|---|---|---|
| | Perplexity | MMLU | Perplexity | MMLU |
| AG-News | 12.757 | 0.510 | 8.022 | 0.533 |
| Emotions | 11.630 | 0.526 | 8.063 | 0.526 |
| DBPedia-14 | 12.230 | 0.507 | 7.903 | 0.535 |

GPT-2, despite being fine-tuned but trained in an autoregressive manner, shows substantially higher vulnerability with major drops in AG-News ($\Delta_{acc} = -0.849$) and DBPedia-14 ($\Delta_{acc} = -0.782$). This increased sensitivity may be attributed to its autoregressive training objective, which potentially leads to more sequential and less redundant concept encodings. The Llama-3.2-3B model, evaluated in a few-shot setting without task-specific training, experiences the most severe degradation across all datasets (often exceeding $-0.90$), suggesting that pre-trained representations without task-specific fine-tuning are more vulnerable to full neuron interventions.

**General LLM Capabilities.** To ensure that our approach does not compromise general model capabilities, we evaluate the impact of range-based masking on language modeling quality and broad knowledge and reasoning performance using the MMLU benchmark. Table 4 presents the comparative performance of neuron masking and activation range masking. Full neuron masking leads to notable increases in perplexity, whereas range masking results in substantially lower perplexity increases. In terms of MMLU accuracy, neuron masking consistently reduces performance across all settings, while range masking preserves or improves performance in most cases, with degradation observed in only one instance.

**Alternative activation interventions.** We also experiment with alternative activation interventions. Details of the alternative approaches are provided in Appendix G. While dampening and mean replacement methods aim to manipulate without moving too far from the original representation, they exhibit limitations when applied to neurons. Specifically, neuron dampening increases perplexity by 2.9–3.7 points and often degrades MMLU accuracy (up to -0.045), whereas range-based dampening confines perplexity increases to 0.5–0.8 points and occasionally improves MMLU (up to +0.035). Similarly, mean replacement leads to substantial degradation when applied to neurons (perplexity increases of 7.4–8.8), while range-based replacement reduces the impact to below 0.7 points.

However, both dampening and mean replacement methods suffer from rigid static suppression or substitution, failing to account for concept-specific activation dynamics. To address this issue, we introduce a novel **adaptive dampening** technique. This method modulates suppression in proportion to each activation's deviation from its class-conditional mean, enabling suppression guided by sampled observations. Adaptive dampening achieves the strongest balance across all metrics: perplexity remains low (0.41–0.61), MMLU is maintained or improved (up to +0.03), and collateral damage to auxiliary concepts is minimized (CAcc drops

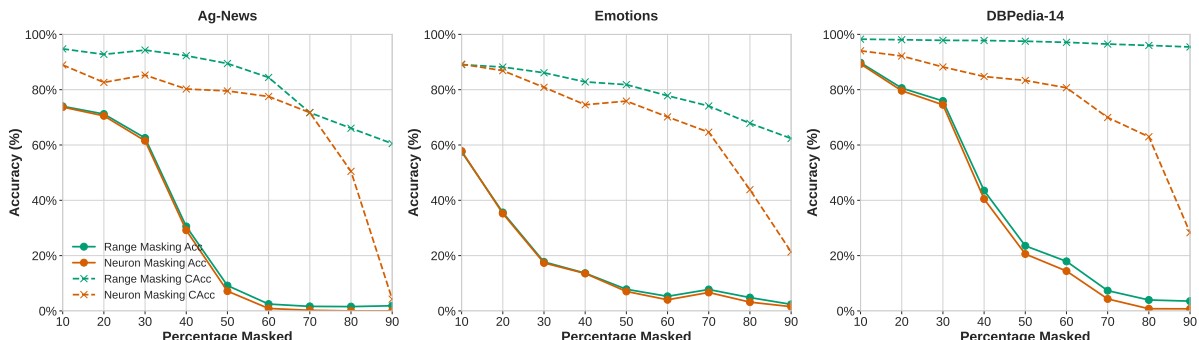

Figure 4: Accuracy on the GPT-2 model as a function of the percentage of masked neurons, comparing activation range-based masking (green) and neuron-level masking (orange).

consistently below -0.3, often under -0.15), outperforming dampening, mean replacement, and zeroing out approaches.

These results demonstrate that precise intervention in specific activation ranges enables substantially more targeted concept manipulation while preserving auxiliary concepts, highlighting how conceptual information is encoded within specific activation patterns rather than isolated to individual neurons, underscoring the importance of activation ranges in capturing neuron-concept relationships.

**Percentage Masking Effect.** As the masking rate increases, the advantage of range-based masking over the neuron-masking baseline becomes increasingly pronounced. Beyond a critical masking level, the baseline exhibits a sharp performance collapse, whereas our method avoids this cliff and degrades smoothly all the way to full masking (100%). This arises from two factors: (1) models have a large number of polysemantic neurons, and higher masking rates increase the chance of ablating them, and (2) blocking/manipulating a higher percentage of the model's neurons creates a significant deviation from the original model's behavior. For low-activation neurons with respect to the concept of interest, discrete neuron masking (i.e., completely masking out a neuron) becomes unreliable, as shown in Figure 4, with a steep performance drop after masking 50% of neurons. Notably, our range-based method offers finer-grained attribution, preserving model behavior under extensive masking. The relatively stable results on auxiliary concepts when using range-based masking at a high percentage of neurons reduce the need to find an optimum threshold for the number of neurons to ablate, which is critical to neuron masking.

Additional results and discussion for activation steering are provided in Appendix M. We find that activation range gating is also helpful in activation steering.

Additional results for Sparse Auto Encoders(SAEs) are provided in Appendix N. We find that activation range masking substantially outperforms full feature masking, ARM produces similar deterioration to full feature masking, while producing lower drops in complementary concepts, and substantially lower perplexity increase.

## 6 Related Work

While we have discussed closely related approaches in § 2, we briefly review additional relevant techniques here. Circuit discovery identifies groups of neurons that jointly encode concepts, providing a structured view of model behavior (Marks et al., 2024; Conmy et al., 2023; Olah et al., 2020). However, extracting circuits is computationally intensive and lacks fine-grained neuron-level attribution. Gradient-based methods attribute predictions to input features by tracking gradients through the network, with integrated gradients (Sundararajan et al., 2017; Dai et al., 2022) being a widely used approach. However, they struggle with polysemanticity, as they do not disentangle overlapping concepts within neurons. Causal analysis methods intervene on internal components to assess their role in encoding concepts. Causal tracing measures the effect of corrupting activations on model performance (Vig et al., 2020; Meng et al., 2022), while causal

mediation analysis quantifies information propagation through neurons (Vig et al., 2020). Although effective, these techniques require costly perturbation experiments. Beyond neuron-level analysis, representation-level methods examine hidden states and their relationship to model outputs and concepts (Veldhoen et al., 2016; Tenney et al., 2019; Liu et al., 2019). Sparse probing (Gurnee et al., 2023b) compresses hidden representations into sparse, interpretable subspaces. While prior work has advanced interpretability, most methods rely on discrete neuron-to-concept mappings, which fail to account for polysemanticity (Sajjad et al., 2022). Our work extends activation-based approaches by characterizing concept-conditioned activation distributions and identifying concept-specific activation ranges that enable more precise attribution and intervention than discrete neuron-to-concept mappings.

While related efforts have analyzed polysemanticity by grouping neuron activation behaviors, they typically rely on clustering heuristics rather than modeling the underlying distributional structure of activations. La Rosa et al. (2023) record activations of vision models across concepts and apply K-means clustering to cluster activations per concept using a fixed number of clusters across settings. Kopf et al. (2025) study polysemanticity by collecting inputs that elicit the highest activations for a neuron, clustering these top-activating inputs, and using an LLM to summarize the resulting clusters as a measure of polysemanticity. In contrast, we study polysemanticity through the distributional analysis of concept-conditioned activations and derive concept-specific activation ranges that facilitate fine-grained, range-based interpretations.

Motivated by the superposition view of representations (Elhage et al., 2022), a complementary line of work builds on sparse dictionary learning (Olshausen & Field, 1997; Lee et al., 2006) and trains sparse autoencoders (SAEs) to learn sparse, monosemantic features from model representations (Yun et al., 2021; Cunningham et al., 2023). By constructing a learned sparse basis, SAEs often yield features that are easier to interpret than those obtained by analyzing the original model representations. NeuronLens is complementary to this line of work. Rather than learning an additional basis, NeuronLens interprets and intervenes on existing model representations by exploiting ranges in concept-conditioned activation distributions, enabling efficient analysis of polysemantic neurons without additional training. Moreover, NeuronLens can complement SAEs by applying activation-range analysis to SAE features that are not completely monosemantic. While SAEs can be powerful, training them can be compute-intensive and poses practical challenges, including dead, duplicate, or missing features and sensitivity to initialization (Bricken et al., 2023; Bussmann et al., 2024; Paulo & Belrose, 2025). Beyond SAEs, our range-based analysis is compatible with broader efforts in interpretation and control, such as representation engineering (Zou et al., 2025) and steering (Tan et al., 2024; Chalnev et al., 2024; Subramani et al., 2022). We view integrating activation-range analysis into these control paradigms as a promising direction for future work.

## 7 Conclusion

In this work, we revisited neuron-level interpretability through the lens of polysemanticity and showed that, despite encoding multiple concepts, individual neurons exhibit predictable activation behavior. Through systematic qualitative and quantitative analysis, we demonstrated that concept-conditioned neuron activations form Gaussian-like distributions with limited overlap. Building on this observation, we introduced NeuronLens, a range-based framework that attributes concepts to activation ranges rather than entire neurons. NeuronLens enables more precise concept manipulation by selectively intervening within these ranges while substantially reducing unintended interference with auxiliary concepts. Our causal evaluations show that range-based masking consistently outperforms full neuron masking and preserves general model capabilities, including broad knowledge and reasoning performance measured by MMLU, as well as language modeling quality.

## 8 Acknowledgment

This work was supported in part by the National Science Foundation (NSF) under grant IIS-2401685 and UNITE Research Priority Area at the University of Kentucky.

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

## A    Limitations

While NeuronLens can disentangle polysemanticity to a degree using identified Gaussian-like distributions, it is unable to completely disentangle concepts encoded in the polysemantic neurons, as concept-specific activation distributions still partially overlap. We also present results primarily from the penultimate layer, and not the intermediate or earlier layers; however, we provide ablations and rationale for this choice in Appendix F

## B    Impact Statement

This work advances neural network interpretability by providing a fine-grained understanding of concept encoding in language models. The proposed NeuronLens framework enables precise control of model behavior, benefiting research in model safety and reliability. While this improved understanding could potentially be misused, the work's theoretical nature and focus on interpretability methods make immediate harmful applications unlikely.

## C    Details of Saliency Methods

**Max Activations  (Frankle & Carbin, 2019).** Max activations extract high neural activations as a saliency ranking metric relying upon the rationale that maximally activating neurons are salient as these neurons play a critical role in controlling the model's output, highlighting their importance for a concept $c$. To identify them, the column-wise mean of absolute neuronal activations in $H_c^l$, $H_c^l$ is defined in §2.3, is computed, given that high negative activations also carry significant signals (Voita et al., 2023). The magnitude of the means is then considered as a ranking for concept $c$.

**Probe Analysis (Dalvi et al., 2019b).** Probe analysis trains a linear classifier on the hidden representations $H_c^l$ to distinguish between concepts. The learned model weights are then utilized as a saliency ranking. This process involves learning a weight matrix $W \in \mathbb{R}^{d \times |c|}$, where $d$ is the hidden dimension and $|c|$ is the number of concept classes. The absolute weight values of each row in the weight matrix are used as a ranking for the importance of each neuron for a given concept. To prevent the emergence of redundant solutions characterized by minimal variations in the weights, the probe is trained using the elastic regularization technique.

**Probeless (Antverg & Belinkov, 2022).** Probeless examines individual neurons, without the need for auxiliary classifiers, using the element-wise difference between mean vectors. The element-wise difference between mean vectors is computed as $r = \sum_{c,c' \in C} |q(c) - q(c')|$, where $r \in \mathbb{R}^d$ and $d$ is the hidden dimension. The final neuron saliency ranking is obtained by sorting $r$ in descending order.

### C.1    Causal validation

To causally validate the aforementioned approaches, we apply erasure; the results for this are provided in Table 6. The results show that degradation of the target concepts is highest when using max activations.

## D    Qualitative and Quantitative Analysis on GPT-2

### D.1    Qualitative

Following similar experimental setup as used in § 4, here we provide the results for the GPT-2 model. Figure 5 and 6 show these visualizations. Figure 5, similar to the Llama, shows that while the activations are Gaussian-like for different concepts, salient neurons demonstrate distinct activation patterns with limited overlap.

In Figure 6, we identify and visualize two distinct types of polysemantic neurons that appear in the salient sets across all classes, when 5% salient set was selected, in the dataset. The first type, exemplified by neuron 480, maintains partially separable activation patterns despite being polysemantic, suggesting some degree of class-specific behavior. In contrast, the second type, represented by neuron 675, exhibits completely

Table 5: Distribution of pairwise overlaps across quartiles (Q1–Q4) for different models and datasets. Q1 includes overlaps <25%, and Q4 includes overlaps >75%. The final row for Llama-3-3.2B reports results for all data aggregated for overlap analysis.

| Model | Dataset | Q1 ($< 25\%$) | Q2 ($25\% - 50\%$) | Q3 ($50\% - 75\%$) | Q4 ($> 75\%$) |
|---|---|---|---|---|---|
| **BERT** | DB-14 | 85.90% | 6.15% | 4.19% | 3.76% |
| | Emotions | 56.11% | 17.87% | 14.18% | 11.83% |
| | AG_News | 52.26% | 20.01% | 14.80% | 12.93% |
| **GPT-2** | DB-14 | 66.70% | 13.53% | 10.39% | 9.38% |
| | Emotions | 21.95% | 25.72% | 26.46% | 25.87% |
| | AG_News | 22.94% | 26.97% | 25.74% | 24.35% |
| **Llama-3-3.2B** | DB-14 | 26.75% | 25.41% | 24.17% | 23.67% |
| | Emotions | 4.41% | 19.72% | 34.13% | 41.74% |
| | AG_News | 17.93% | 26.46% | 27.53% | 28.09% |
| | All Data | 43.38% | 20.15% | 18.46% | 18.01% |

Table 6: Performance drops relative to Baseline configuration (i.e., unaltered model's performance) for three techniques: Probeless, Probe, and Max. All values show the difference from Base values. Results are for *Emotions* dataset on the GPT-2 and Llama-3-3.2B model using 30% salient neurons of each method. Metrics are detailed in § 5.3.

| Model | Probeless | | | | Probe | | | | Max | | | |
|---|---|---|---|---|---|---|---|---|---|---|---|---|
| | $\Delta$**Acc** | $\Delta$**Conf** | $\Delta$**CAcc** | $\Delta$**CConf** | $\Delta$**Acc** | $\Delta$**Conf** | $\Delta$**CAcc** | $\Delta$**CConf** | $\Delta$**Acc** | $\Delta$**Conf** | $\Delta$**CAcc** | $\Delta$**CConf** |
| **GPT-2** | -0.524 | -0.510 | -0.086 | -0.086 | -0.052 | -0.036 | **-0.018** | **-0.049** | **-0.735** | **-0.739** | -0.103 | -0.103 |
| **Llama** | -0.751 | -0.292 | -0.751 | -0.272 | -0.177 | -0.145 | **-0.169** | **-0.096** | **-0.805** | **-0.500** | -0.511 | -0.399 |

overlapping activation patterns across all classes, making it hard to disentangle. To further investigate this phenomenon, Figure 7 presents a broader analysis of neurons from the polysemantic subset, identified using a 5% saliency threshold (top 5% salient neurons selected for a concept).

### D.2 Quantitative

Skewness and Kurtosis, and practical normality results on *GPT-2* model are provided in Table 7. The results are similar to those observed on the Llama model in the main text.

Table 5 provides results regarding neuron activation overlap for various concepts using AUROC, considering all possible pairwise combinations of concepts within datasets.

## E Qualitative Text Analysis

After recording the activations and calculating distributions, we calculate the activation range (as defined in § 5.1) for each class in the dataset under consideration. We then extract 10 text examples at each of three points along the computed activation range for each concept: the lower bound (minimum of the activation range), the midpoint (mean of the activation range), and the upper bound (maximum of the activation range).

We then pass these text examples to an LLM judge (Gemini Pro) along with class labels and boundary position information. The LLM judge then extracts the underlying semantics shared in the examples from one of the labels of the dataset, along with highlighted keywords. We present 3 representative examples from each set for one selected class, along with LLM highlighted keywords and semantics in the Table 2, the full relevant texts for the table are provided in Table 8.

Table 7: Skewness, kurtosis, and Kolmogorov-Smirnov test results across various datasets. *GPT-2* model

| Dataset | Skewness | Kurtosis | KS-Test |
|---|---|---|---|
| stanfordnlp/imdb | 0.0014 | 3.6639 | 1.0000 |
| fancyzhx/dbpedia_14 | -0.0007 | 3.9360 | 0.9782 |
| dair-ai/emotion | 0.0015 | 3.0198 | 0.9446 |
| fancyzhx/ag_news | -0.0013 | 3.2060 | 0.9918 |
| stanfordnlp/sst2 | -0.0083 | 3.2038 | 1.0000 |

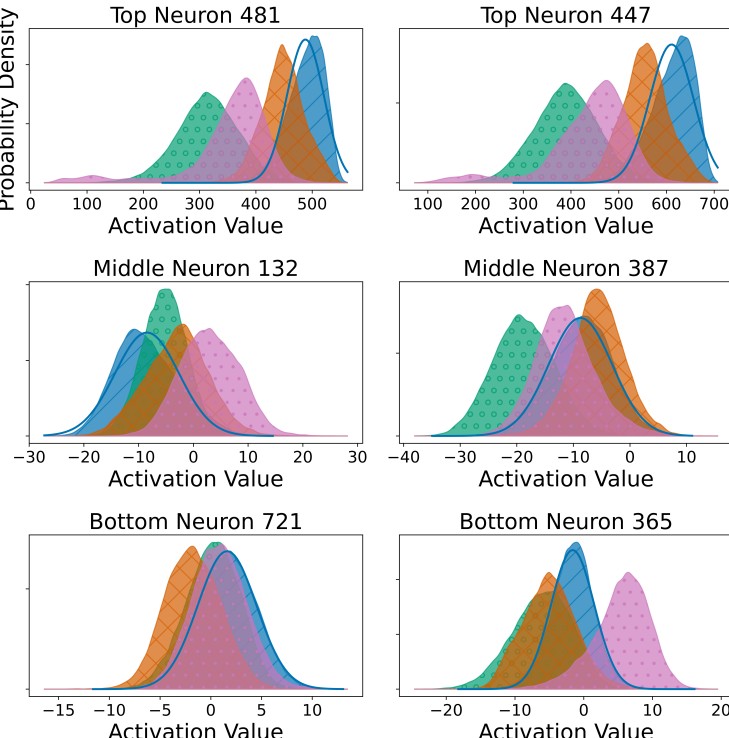

Figure 5: Neuronal Activation Patterns of six neurons on *AG-News* dataset *class 1*. Neurons 481 and 447 are the highest activating neurons, neurons 132 and 387 are middle-ranked neurons, and neurons 721 and 365 are the lowest activating neurons.

# F    Layer Ablation

## F.1    Statistical Results

We analyze concept level activation distributions across all 12 layers of GPT-2, measuring kurtosis (where a value of 3.0 indicates a Gaussian distribution), skewness (where 0 indicates symmetry), and practical normality in Table 9:

These results show that kurtosis values converge toward 3.0 (the Gaussian ideal) as layers progress, skewness values remain near zero across all layers, and practical normality scores are close to 1 across all layers. Importantly, if the activations were not clustered into continuous intervals and were in disconnected islands of activations, these would be reflected in the score for the practical normality and other statistical metrics.

## F.2    Qualitative Results

We expanded our visualization approach, shown in figs. 8 to 19) to all layers in the model. The visualizations demonstrate an interesting progression: while all layers exhibit Gaussian-like distributions on the class level, class concepts aren't separated in the activation spectrum Gaussians of the early layers. This aligns with the

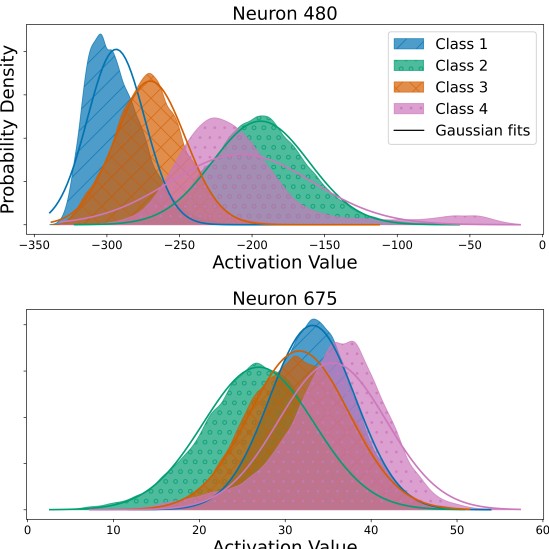

Figure 6: Neuronal Activation Patterns comparison of neurons 480 and 675. The plots show class-specific activity patterns with fitted Gaussian curves. Both neurons were part of salient set of all classes when top 5% salient selected on *AG-News* dataset.

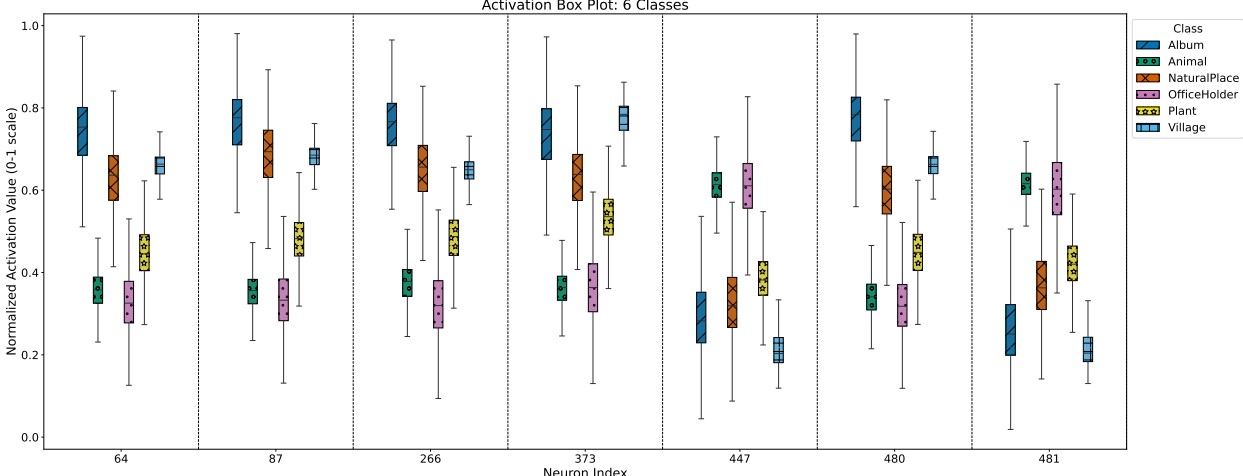

Figure 7: Box plot of neural activation of 11 polysemantic neurons (i.e: neurons in the salient group for all classes, percentage selected: 5% top salient) for 4 randomly selected classes out of 14 classes of *DBPedia-14* dataset.

understanding that lower layers capture more basic features rather than high-level semantic features like class. However, distinct concept-level Gaussian distributions begin forming as early as layers 5-6, becoming increasingly separable in deeper layers.

## F.3 Masking Results

In Table 10 and Table 11 we provide results of applying both approaches on all layers of *GPT-2* model on *Emotions* dataset. From the results we can see that: Early layers (1-3) show highly variable and often severe impacts: Layer 1 exhibits minimal effects ($\Delta Acc = -0.113$, $\Delta CAcc = -0.064$), while Layers 2-3 show extreme degradation ($\Delta Acc \approx -0.7$, $\Delta CAcc > -0.5$). Middle layers (4-8) demonstrate inconsistent behavior with high variance in impacts. Layer 12, however, achieves an optimal balance: it maintains substantial primary task

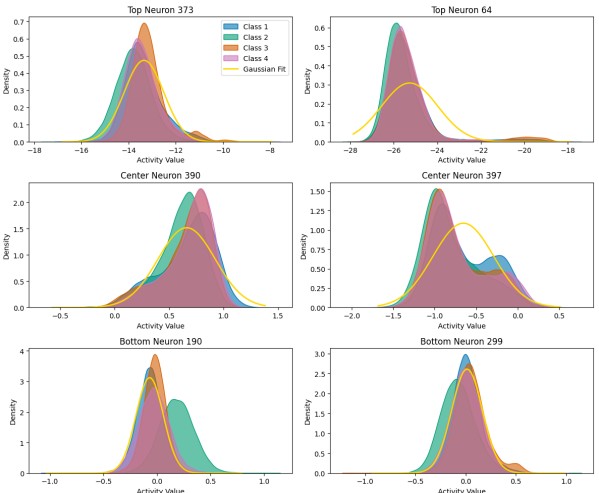

Figure 8: Neuronal Activation Patterns of six neurons on *AG-News* dataset. Layer 1

Figure 9: Neuronal Activation Patterns of six neurons on *AG-News* dataset. Layer 2

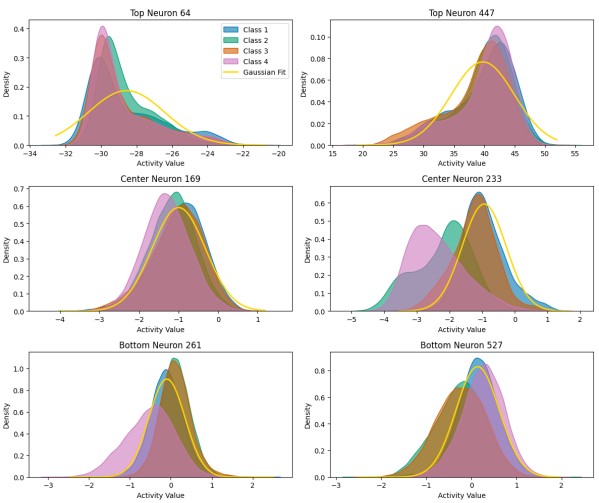

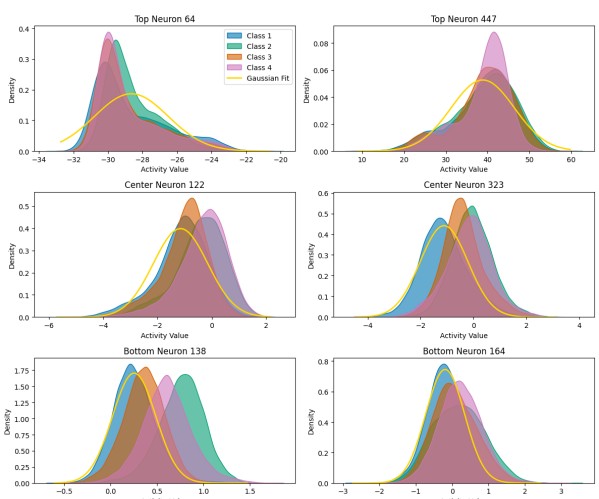

Figure 10: Neuronal Activation Patterns of six neurons on *AG-News* dataset. Layer 3

Figure 11: Neuronal Activation Patterns of six neurons on *AG-News* dataset. Layer 4

impact ($\Delta Acc = -0.571$) while minimizing auxiliary concept interference ($\Delta CAcc = -0.060$). This pattern holds true for both neuron masking and range masking techniques, with range masking showing slightly better preservation of auxiliary concepts ($\Delta CAcc = -0.045$). The mid-range primary task degradation combined with minimal auxiliary impact makes Layer 12 the most suitable for targeted interventions, offering better control and specificity compared to earlier layers.

## G  Alternative Activation Interventions

In the main text, we primarily presented results using a "zeroing out" strategy for neuron manipulation. This approach was chosen to compare neuron manipulation against range-based manipulation. However, zeroing out is considered a suboptimal strategy (Suau et al., 2024). The primary concern with standard zeroing-out approaches is that they distort the activation distribution significantly, diverging from that of the original model. However, our range-based method selectively zeroes out only a narrow slice of the activation spectrum, thereby mitigating the adverse effects associated with hard erasure.

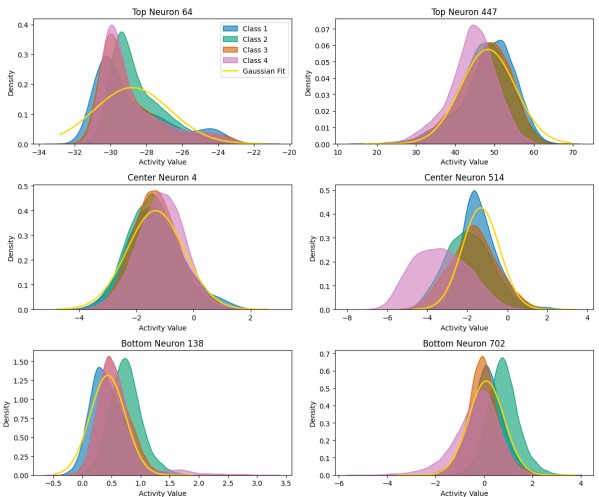

Figure 12: Neuronal Activation Patterns of six neurons on *AG-News* dataset. Layer 5

Figure 13: Neuronal Activation Patterns of six neurons on *AG-News* dataset. Layer 6

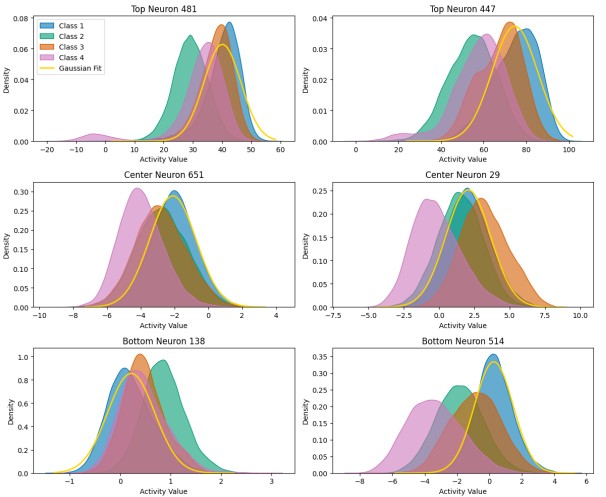

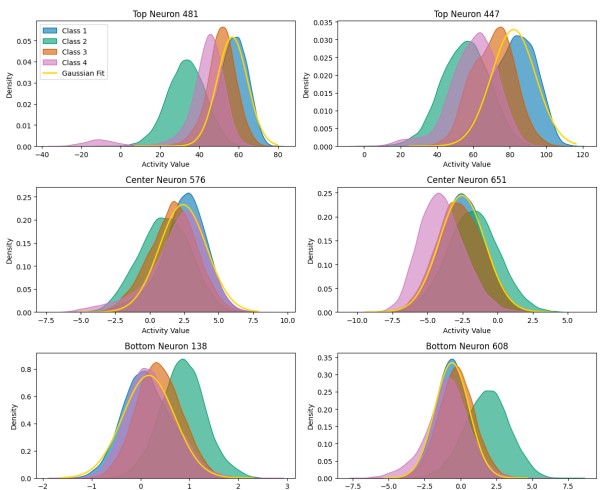

Figure 14: Neuronal Activation Patterns of six neurons on *AG-News* dataset. Layer 7

Figure 15: Neuronal Activation Patterns of six neurons on *AG-News* dataset. Layer 8

In this section, we explore alternative, more optimized strategies for concept removal. We also introduce a novel range-based scaling strategy that has demonstrated superior results.

Below, we explore various activation intervention strategies, comparing traditional neuron-level approaches with the nuanced range-based technique. Our comprehensive evaluation reveals that range-based manipulations consistently outperform neuron interventions across multiple metrics, with significantly less disruption to the model's general capabilities.

Among all techniques examined, our novel adaptive dampening approach emerges as the most effective, maintaining targeted concept suppression while minimizing collateral impact on auxiliary concepts and preserving overall language modelling capabilities. This pattern holds true across different intervention methods including zeroing out, dampening, and mean replacement strategies.

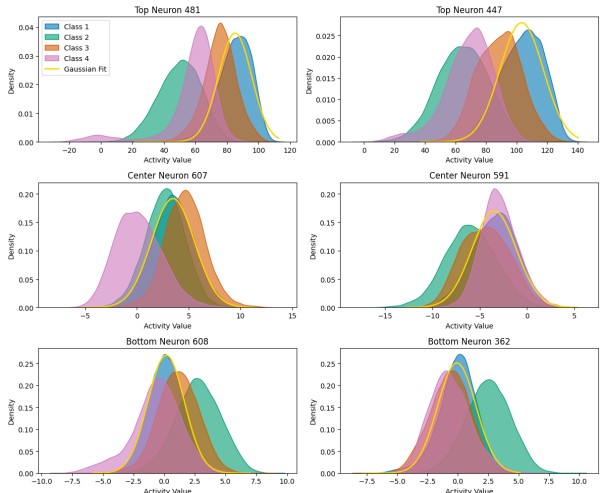

Figure 16: Neuronal Activation Patterns of six neurons on *AG-News* dataset. Layer 9

Figure 17: Neuronal Activation Patterns of six neurons on *AG-News* dataset. Layer 10

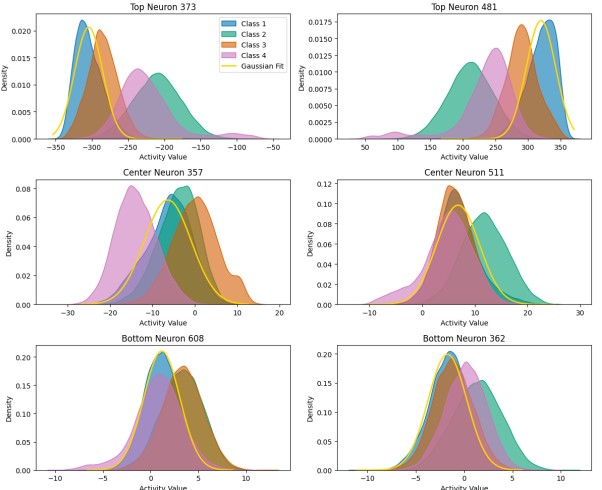

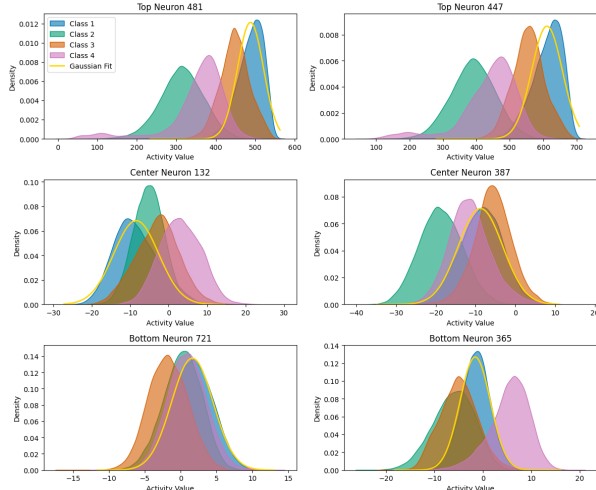

Figure 18: Neuronal Activation Patterns of six neurons on *AG-News* dataset. Layer 11

Figure 19: Neuronal Activation Patterns of six neurons on *AG-News* dataset. Layer 12

## G.1 Dampening

In their work, Suau et al. (2024) propose using a dampening function rather than setting neuron activations to zero outright. This approach, referred to as DAMP, corresponds to a specific choice of the intervention function $\phi(x) = \alpha x$, where $0 \leq \alpha \leq 1$. In this formulation, the activations of selected neurons are scaled down by a factor $\alpha$ instead of being completely suppressed. Here, $x$ represents neuron activation. The rationale behind dampening is that a fixed intervention (like zeroing out) can disrupt the LLM's inference dynamics, especially when a large number of neurons ($k$) are involved, thereby limiting its effectiveness. Dampening offers a less destructive intervention by allowing contextual signals to continue passing through the network. This, in turn, permits intervention on a larger set of expert neurons, potentially achieving stronger mitigation of the targeted concept.

Table 12 presents a comparative analysis of two intervention strategies, neuron masking and activation range masking, when employing the Dampening technique with $\alpha = 0.5$. The evaluation spans 14 classes and

utilizes the metrics: accuracy (Acc), confidence (Conf), class-wise accuracy (CAcc), class-wise confidence (CConf), alterations in perplexity (PPL), and MMLU score.

A consistent trend emerges across the primary metrics (Acc, Conf, CAcc, and CConf), where activation range masking demonstrates superior performance over neuron masking. Interventions based on activation ranges lead to a notably smaller decline in the accuracy and confidence associated with auxiliary concepts. For example, in Class 3, while neuron masking results in an accuracy drop of -0.974 in the targeted class and auxiliary class accuracy decrease of -0.411, activation range masking, despite a comparable accuracy reduction in the targeted class (-0.970), shows a less severe impact on auxiliary class accuracy (-0.283). This pattern of activation range masking better preserves auxiliary class performance, is evident across all evaluated classes.

Examining the broader effects on language modeling capabilities reveals significant distinctions between the two approaches. Neuron masking results in a considerable rise in perplexity (PPL), with increases ranging from **+2.891 to +3.732** across all the classes. Furthermore, it tends to cause more pronounced negative shifts in MMLU scores, reaching as low as -0.045. Conversely, activation range masking results in substantially smaller increments in perplexity, falling within the **+0.546 to +0.810** range, and frequently results in improved or minimally altered MMLU scores, with gains up to +0.035.

### G.2 Mean Replacement

Another approach of activation replacement discussed in the literature(Suau et al., 2021) is replacing it with the mean activation value. We provide the results for this type of replacement in Tables 13 14 15.

The mean replacement strategy corresponds to setting the intervention function to $\phi(x) = \mu$, where $\mu$ is the mean activation of the neuron $x$ computed over a general next-token prediction task on the Wikipedia (Foundation).

In Tables 15, we assess the effect of mean replacement using both neuron masking and activation range masking on the DB-Pedia dataset. In every class, neuron masking results in more severe degradation than range-based masking across all auxiliary and general metrics.

Across metrics (Acc, Conf, CAcc, and CConf), activation range masking consistently outperforms neuron masking. The degradation in accuracy and confidence of auxiliary concepts is significantly lower under range-based interventions. For instance, in Class 3, neuron masking causes a drop of -1.000 in Acc and -0.766 in CAcc, whereas activation range masking yields a similar Acc drop (-1.000) but a substantially smaller decline in CAcc (-0.538). A similar pattern repeats across all classes; for example, in Class 0, neuron masking results in CAcc of -0.685 while activation range masking yields -0.551. In Class 7, neuron masking shows a CAcc of -0.491 compared to -0.267 for activation range masking.

Beyond auxiliary class performance, we observe substantial differences in how the two masking methods affect general language modelling capabilities. Neuron masking leads to a large increase in perplexity (PPL), ranging from **+7.413 to +8.791** which is catastrophic, across classes, and induces more negative shifts in MMLU scores (as low as -0.035 for Class 9, and also for Class 0 with -0.025, Class 1 with -0.030, Class 10 with -0.020, and Class 13 with -0.025). In contrast, activation range masking results in substantially smaller increases in perplexity (+0.397 to +0.687) and often yields improved or near-zero changes in MMLU scores (up to +0.020 for Class 6, and several positive values like +0.015 for Class 1, Class 8, and Class 9).

### G.3 Adaptive Dampening

We propose a novel replacement method in which the intervention function $\phi(x)$ applies *runtime-controlled dampening* based on the distance of the observed activation $x$ from the center of a predefined activation range. Specifically, the dampening factor $a(x)$ is linearly scaled according to the distance of $x$ from the mean $\mu$ of the neuron's activation distribution, within the range $[\mu - 2.5\sigma, \mu + 2.5\sigma]$.

Let $\beta \in [0, 1]$ denote the maximum dampening factor applied at the range boundaries. Then:

$$a(x) = \beta \cdot \frac{|x - \mu|}{2.5\sigma}, \quad \text{and} \quad \phi(x) = a(x) \cdot x.$$

This ensures that when $x = \mu$ (the center of the range), $a(x) = 0$ and the activation is fully suppressed via $\phi(x) = 0$. At the boundaries ($x = \mu \pm 2.5\sigma$), $a(x) = \beta$, and the activation is minimally dampened. Values within the range are scaled proportionally based on their normalized distance from the mean. This adaptive dampening mechanism suppresses values near the mean while preserving those closer to the range edges.

The dampening factor $\beta$ can be optimized for different neurons based on the concept information that neuron provides. For this work, we use $\beta = 0.5$ across all neurons.

In Table 16 we evaluate the adaptive dampening variant of the replacement function. This approach outperforms both neuron masking and static activation masking across all metrics.

In auxiliary class metrics, adaptive dampening yields much smaller degradation. Auxilary class accuracy (CAcc) and confidence (CConf) show significantly reduced drops compared to other methods. For example, in Class 0, CAcc drops only $-0.215$ compared to $-0.685$ under neuron masking and $-0.551$ under hard activation masking. The effect is consistent across classes, with most CAcc and CConf drops staying well below $-0.3$, and in many cases below $-0.15$.

Language modeling metrics show this approach to be exceptionally efficient. Perplexity increases are minimal, remaining within $+0.408$ to $+0.609$, substantially lower than all hard-masking variants. MMLU deltas also stay close to zero, with several classes showing improvement (e.g., Class 8: $+0.030$, Class 4: $+0.015$). Notably, no class suffers significant MMLU degradation.

## H    Training Details

For BERT, DistilBERT, and Llama, we utilize pretrained models. Since BERT, and DistilBert are not inherently trained as a conversational agent, we use top-performing fine-tuned models from the Hugging Face repository. For the Llama model, few-shot prompt completion is employed to predict class labels. This involves providing a small number of training samples from the dataset to guide the model's predictions.

For GPT-2, we fine-tune the pretrained model across all datasets for three epochs. The input sequence is constructed by concatenating the text with a <sep> token, followed by the class label, and ending with an <eos> token. During training, the loss is back-propagated only for the class label token, while all other tokens are assigned a skip label (-100). Additionally, all class labels are added to the model's dictionary as special single-token entries.

In the case of BERT-based models, record the activation of the CLS token. In the case of GPT-2 and Llama models, we record the last token output when the class token is being predicted. The intervention is applied to the appropriate token on the residual stream.

For trained models (BERT, DistilBERT, and GPT-2), a higher proportion of neurons (up to 50%) can be ablated with a relatively minor impact on primary task performance and minimal interference with auxiliary concepts. This suggests substantial neuronal redundancy, wherein multiple neurons appear to encode overlapping features.

**Dataset Preprocessing for Llama** For Llama we process whole datasets in few-shot settings and only curate 2000 samples per class, where the model prediction was correct.

### H.1    Computation Details

All experiments, including activation extraction and interventions on large language models (LLMs), were conducted using an NVIDIA RTX 3090 GPU equipped with 24GB of VRAM. 64GB RAM.

## I    Hyperparameter Ablation

The results for ablating $\tau$ on GPT-2 model using the AG-News dataset are shown in Table 17. For the target concept, values $0.3 - 2.4$ show decreasing accuracy/confidence, stabilizing at $\tau = 2.4$ (accuracy 0.6126). Beyond 2.4, negligible additional degradation occurs, indicating we've captured the complete target concept

activation range. Importantly, while target performance stabilizes after $\tau = 2.4$, auxiliary task performance declines after $\tau = 2.7$. Complement accuracy stays above 0.93 until then before dropping to 0.8795 at $\tau = 4.5$. This aligns with normal distribution properties where 95-99% of values fall within $\pm 2.5$ standard deviations.

## J  Class Wise Results

Here we provide the complete results for the selected models for all datasets. IMDB (Table 24), SST2 (Table 25), AG-News (Table 23), Emotions (Table 26) and DBpedia-14 (Table 27)

## K  Fine Grained Concepts

We evaluate the extent to which our interventions suppress the model's ability to predict finer-grained concepts, including pronouns, delimiters (brackets), terminal punctuation, and subject-verb agreement (Chauhan et al., 2025; Gurnee et al., 2023a; Rai et al., 2025; Miller et al., 2025; Marks et al.). We use a set of pronouns spanning masculine, feminine, and neutral forms: he/his, she/her, and they/them. For terminal punctuation, we analyze periods (.), question marks (?), and exclamation marks (!). For delimiters, we consider the opening braces {, (, and [, and for subject–verb agreement, we analyze both singular and plural verb forms, specifically do/does and go/goes.

For each concept, we curate a dataset from WikiText by extracting 500 samples and manually verifying the data. We use 400 samples to rank neurons and estimate activation ranges. The remaining 100 samples form the held-out test set. We initially evaluate ranking methods as discussed in the main text. Our initial experiments suggested that probe-based ranking was not effective. To reduce compute overhead, we analyze max and probeless rankings on pronoun concepts. We find that overall results are similar, as shown in Table 18, but the max ranking yields a higher perplexity score. We therefore adopt probeless ranking for all reported results on fine-grained concepts.

The results for pronouns, delimiters (brackets), terminal punctuation, and subject-verb agreement are shown in Table 19 20 21 22, respectively. The results mirror the findings from the main-text experiments: the degradation of the chosen concept is comparable to full neuron ablation, while the auxiliary concepts are better preserved. This reduced collateral impact is reflected in the lower perplexity, indicating fewer unintended side effects.

## L  Range Masking Algorithm

Algorithm 1 outlines the overall computation pipeline.

## M  Model Activation Steering:

Here we compare our method, NeuronLens, with model steering on the Llama 3 3.2B. Steering uses a dataset in which the target concept is present, along with an auxiliary dataset in which the concept is absent (Chen et al., 2025; Pai et al., 2025). In our setting, the intervened concept class serves as the "presence" dataset, while all other concept classes form the "absence" dataset. Following the recommendations of Chen et al. (2025), we apply steering at the model's mid-layer (layer 14). To apply steering, a difference-in-means vector $r$ is constructed using the average activations of targeted and auxiliary concepts. To apply steering at inference time, vector $r$ is scaled by a scaler $a$, and added to the residual stream. In this setting, we evaluated different values of $a$ and found $a = -2$ to be most precise in target concept removal.

$$x' \leftarrow x + a * r \tag{3}$$

In comparison, we apply range-based removal using a conservative setting of $\tau = 1$ across the full representation vector. Results showing a comparison for activation steering and ARM are reported in Table 28. The results suggest that steering some concepts causes only minimal degradation in the targeted behavior(Namely class 0), which indicates that a clear steering direction does not exist for this concept. For other concepts, performance

---

**Algorithm 1** Range-Based Masking (Activation Range Masking)

---

**Require:** Model $M$; labeled dataset $D = \{(x_i, y_i)\}$; target layer index $l$; concept $c$; top fraction $p$; range width $\tau$; range-gated operator $\phi(\cdot)$; ranking function $R(j, H_c^l, H_{\neg c}^l) \to \mathbb{R}$.
1: Let $[d] = \{1, \ldots, d\}$ denote neuron indices.
**Ensure:** Modified model $M'$ where layer $l$ applies range-based masking in its forward.
2: **Offline: collect concept-conditioned activations**
3: $D_c \leftarrow \{(x_i, y_i) \in D : y_i = c\}$
4: $D_{\neg c} \leftarrow \{(x_i, y_i) \in D : y_i \neq c\}$
5: Run $M$ on $D_c$ and store hidden vectors at layer $l$:
6: $\quad H_c^l \leftarrow \{h^l(x_i) : (x_i, y_i) \in D_c\}$
7: Run $M$ on $D_{\neg c}$ and store hidden vectors at layer $l$:
8: $\quad H_{\neg c}^l \leftarrow \{h^l(x_i) : (x_i, y_i) \in D_{\neg c}\}$
9: **Offline: rank and select neurons**
10: **for** $j \in [d]$ **do**
11: $\quad s_j^c \leftarrow R(j, H_c^l, H_{\neg c}^l)$
12: **end for**
13: $S_c \leftarrow$ indices of top-$p$ fraction by $s_j^c$
14: **Offline: compute attribution ranges**
15: **for all** $j \in S_c$ **do**
16: $\quad \mu_j \leftarrow \frac{1}{|H_c^l|} \sum_{h \in H_c^l} h_j$
17: $\quad \sigma_j \leftarrow \sqrt{\frac{1}{|H_c^l|} \sum_{h \in H_c^l} (h_j - \mu_j)^2}$
18: $\quad AR(l, j, c) \leftarrow [\mu_j - \tau\sigma_j, \; \mu_j + \tau\sigma_j]$
19: **end for**
20: **Update layer $l$: apply range-gated intervention in forward**
21: For any input $x$, compute $h^l \leftarrow \text{Layer}_l(x)$
22: **for all** $j \in S_c$ **do**
23: $\quad$ **if** $h_j^l(x) \in AR(l, j, c)$ **then**
24: $\quad\quad h_j^l(x) \leftarrow \phi\big(h_j^l(x)\big)$
25: $\quad$ **end if**
26: **end for**
27: Return $h^l(x)$ and continue the forward pass of $M$ normally.

---

is comparable to activation range masking; however, activation steering significantly increases perplexity scores.

### M.1 Directional Ablation:

We also compare `NeuronLens` with directional ablation as described in (Arditi et al., 2024).

$$x' \leftarrow x - \hat{r}\hat{r}^T x \tag{4}$$

We apply projection removal to only one layer, layer 14. Table 30 reports the results. We find that directional ablation fails in this setting, eliminating both target and auxiliary classes and resulting in a very high perplexity.

### M.2 Range Gated Steering:

To access the effect of combining `NeuronLens` with activation steering, we perform a preliminary experiment of using activation steering instead of zero or mean ablation in our range-based framework (i.e, only steering components if their activation falls within the activation range of the target concept). In this setting, for the calculation of the mean values targeted and auxiliary concepts, we utilize the last contextualized token

instead of the average token representation (which is the default setting for activation steering). We compare this approach with activation steering.

We present the results of the experiment in Table 29. The results show that base activation steering is able to precisely manipulate the targeted concept, but at a substantial cost of perplexity. Our activation range-based steering (Activation Steering) produces similar results but bounds the perplexity to a much lower value.

## N    Comparison with Sparse Autoencoders

Sparse Autoencoders (SAEs) learn to represent a concept using a sparse linear combination of neuron activations, where neurons are encouraged to be monosemantic in nature. NeuronLens can be seen as unrolling a neuron activation to identify monosemantic activation ranges, each belonging to a unique concept, with limited overlaps. NeuronLens and SAEs can be viewed as complementary methods: SAEs may serve as a first step toward disentangling polysemanticity, while NeuronLens can further provide fine-grained monosemantic ranges on top of the SAE representations. In this section, we conduct a preliminary experiment on concept removal in SAE features, comparing full feature blocking with activation-range masking applied to SAE features.

### N.1    Experimental Setup

Since pretrained SAEs are not available for the models tested in our main experiments, we utilize pretrained GemmaScope SAEs on Gemma-2-2B. Experiments are conducted on our largest tested dataset (DBPedia) for two variants of SAEs:

- Gemma Scope (Width_16k / l0_285)

- Gemma Scope (Width_65k / l0_197)

We initially record the activated SAE features using samples from the training set, and then filter these based on their frequency of activation. A feature is only considered for ablation if it activates for at least 10% of the training set for a given concept. From this subset, we select the top 20% highest-activating features and apply either full feature masking or activation-range masking to test causality. Details on the selected features in both settings are provided in Table 31.

### N.2    Results

Table 32 summarizes the experimental results. Compared to feature masking, activation-range masking produces a smaller reduction in complementary/auxiliary concepts and significantly lower perplexity scores, while achieving a drop on the targeted concepts that is comparable to full feature masking.

These patterns suggest that SAE features are not strictly monosemantic, and they still contain neurons that capture mixtures of related concepts. While NeuronLens is effective in interpreting original polysemantic neurons, it can also be used to disentangle SAE-learned representations, achieving more precise concept removal with substantially lower impact on overall model fluency.

## O    Dataset Details

In Table 33, we provide samples from the benchmarking datasets used in the main paper. In Table 34, we provide samples from the finer-grained concepts tested in § K.

Table 8: Full text for samples shown in Table2 for Neuron 1525 (Sci/Tech) and Neuron 2587 (Business).

| Neuron | Point | Boundary Affinity | Representative Examples |
|---|---|---|---|
| 1525 | Lower Boundary | World / Sports | "Gene Tweaking Turns Couch Potato Mice Into Racers Altering a single gene turned ordinary mice into marathon racers that could run for hours and eat huge amounts of food without getting fat, a team of researchers reported on Monday." 

 "Insecure elections marching ever closer Friday's St. Louis Post-Dispatch reports on a controversial decision by Missouri's Secretary of State: the state of Missouri will be allowing soldiers stationed overseas to cast ballots via e-mail. Their absentee ballots will be scanned and converted to PDF files, which will be emailed to the Defense Department, printed out, and then faxed to Missouri. I'm in favor of helping soldiers vote; this is a democracy, everyone should be able to vote. Yet I'm deeply skeptical of this proposal, for two reasons: The plan depends on e-mailed ballots being printed out and faxed by the Defense Department but does not provide any safeguards against soldiers being sanctioned for how they have voted; The transmission method is inherently technically insecure " 

 "IBM's new supercomputer breaks the world's fastest computer's ... Technology India: London, Nov 8 : A new supercomputer being constructed by IBM has broken all supercomputing records after demonstrating double the power of the long-reigning supercomputing champion, NEC's Earth Simulator, based at Yokohama, Japan." |
| | Mean | Sci/Tech | "Sony, IBM, Toshiba say powerful chip to start production in 2005 SAN JOSE, Calif. A long-awaited microprocessor developed by IBM, Sony, and Toshiba will go into production next year and start appearing in video game consoles and high-definition T-Vs in 2006." 

 "AMD Gives Details on Dual-Core Opteron Advanced Micro Devices has given out more details on its fortcoming dual-core microprocessor chip. The Opteron-based design is said to be 30-55 percent faster than AMD #39;s single-core chips, but it will fit in existing server designs." 

 "Arm reveals Neon multimedia extension technology Microprocessor designer Arm Ltd. has developed a new multimedia technology called Neon that will help improve the performance of mobile electronics devices that process multiple tasks, the company said Monday." |
| | Upper Boundary | Business | "iPass Introduces New Flat-Rate Pricing Plans for US Wi-Fi Hotspot ... REDWOOD SHORES, Calif., Nov. 17 – iPass Inc. today announced new monthly and annual flat-rate subscription plans for use of the Company's US Wi-Fi connectivity." 

 "NTT DoCoMo Rises on TSE on Reported Deal with Motorola Tokyo, Aug. 23 (Jiji Press)–NTT DoCoMo firmed on the Tokyo Stock Exchange Monday morning following a media report that it will start procuring mobile phone handsets made by Motorola Inc." 

 "Update: Omnipod beefs up instant messaging service Omnipod, which provides hosted instant message (IM) services to companies of all sizes, is preparing several enhancements to its platform, including the additions of a Web-based client, a telephony component and a persistent-chat feature, Omnipod's chief executive officer said." |
| 2587 | Lower Boundary | World / Sports | "Put Me in, Coach! Coach joins the S P 500, and others stand to benefit from the leather in the weather." 

 "London Stock Exchange eyes Asia HQ The London Stock Exchange plans to set up an Asia headquarters in Hong Kong to tap the growing number of mainland corporates eyeing listings abroad, a local newspaper reports." 

 "Halliburton closes higher on Army's decision to pay DALLAS (CBS.MW) – Halliburton's shares closed higher Wednesday after the Army Materiel Command reversed its decision to withhold 15 percent of its future payments to the company under a contract to supply and support US troops in Iraq." |
| | Mean | Business | "UBS Buys Schwab Unit for $265 Mln GENEVA/NEW YORK (Reuters) - Swiss-based banking giant UBS AG has agreed to buy Charles Schwab Corp.'s capital markets unit for $265 million in cash, making UBS a leading player on the U.S. Nasdaq exchange, the companies said on Tuesday." 

 "Applied Materials Applied Materials (AMAT: news, chart, profile) shares were off two cents to $16.05 in trading before the bell Wednesday and had wavered around break-even in late trading Tuesday after the results were announced." 

 "Brown-Forman Earnings Jump 67 Percent Brown-Forman Corp. , which sells products ranging from Jack Daniels whiskey to Lenox china, on Thursday posted a better-than-expected 67 percent jump in earnings as aggressive marketing boosted sales of premium spirits and new wines." |
| | Upper Boundary | Sci/Tech | "Arm hands over $910m for US chip firm Arm is to pay $910m (504m) in cash and shares for Artisan, the US-based transistor-level designer for systems-on-a-chip. Arm chairman Sir Robin Saxby said in a conference call: quot;This will be a combination " 

 "Oracle sales rise on database demand com September 14, 2004, 2:26 PM PT. This fourth priority's main focus has been improving or obtaining CRM and ERP software for the past year and a half." 

 "Coming: IT that adapts to users' requirements The march of information technology into the workplace has been greeted with a mix of awe and resistance. For all their promise of productivity gains, computers, business software, and telecommunications gear have disrupted processes at the core of a company's identity." |

Table 9: Statistical analysis of different layers showing skewness, kurtosis, and Kolmogorov- Smirnov test results. *GPT2* model. *AG-News Dataset*

| Layer | Kurtosis | Skewness | Practical Normality |
|-------|----------|----------|---------------------|
| 1  | 3.9314 | 0.0430  | 0.7913 |
| 2  | 3.7622 | -0.0091 | 0.9525 |
| 3  | 3.4109 | -0.0143 | 0.9870 |
| 4  | 3.5582 | -0.0073 | 0.9801 |
| 5  | 3.6145 | 0.0051  | 0.9730 |
| 6  | 3.5318 | 0.0086  | 0.9769 |
| 7  | 3.3461 | 0.0083  | 0.9880 |
| 8  | 3.2763 | 0.0037  | 0.9870 |
| 9  | 3.2267 | 0.0039  | 0.9860 |
| 10 | 3.2057 | 0.0029  | 0.9899 |
| 11 | 3.2105 | -0.0002 | 0.9912 |
| 12 | 3.2061 | -0.0014 | 0.9919 |

Table 10: Evaluation of layer selection on *GPT-2* model on the *Emotions* dataset using neuron and range masking techniques. 20% Neurons selected. Here, **Acc** represents class accuracy, **Conf** denotes class prediction probability, and **CAcc** and **CConf** refer to average accuracy and average class prediction probability across other classes, respectively. The *Base Values* indicate the baseline model performance, while *Activation Range Masking* and *Neuron Masking* show deviations from the baseline performance.

| Layer | Class | Base Values | | | | Neuron Masking | | | | Activation Range Masking | | | |
|-------|-------|------|------|------|-------|----------------|----------------|-----------------|-----------------|--------------------------|----------------|-----------------|-----------------|
| | | Acc | Conf | CAcc | CConf | $\Delta$Acc | $\Delta$Conf | $\Delta$CAcc | $\Delta$CConf | $\Delta$Acc | $\Delta$Conf | $\Delta$CAcc | $\Delta$CConf |
| 1 | Class 0 | 0.970 | 0.957 | 0.915 | 0.904 | -0.029 | -0.071 | -0.074 | -0.100 | 0.006  | 0.002  | -0.004 | -0.005 |
|   | Class 1 | 0.933 | 0.932 | 0.931 | 0.913 | -0.011 | -0.056 | -0.090 | -0.116 | 0.001  | -0.003 | -0.004 | -0.004 |
|   | Class 2 | 0.901 | 0.865 | 0.934 | 0.924 | -0.206 | -0.195 | -0.052 | -0.092 | -0.019 | -0.015 | -0.001 | -0.002 |
|   | Class 3 | 0.926 | 0.924 | 0.932 | 0.919 | -0.128 | -0.152 | -0.051 | -0.090 | -0.004 | -0.005 | -0.001 | -0.002 |
|   | Class 4 | 0.885 | 0.867 | 0.938 | 0.927 | -0.055 | -0.084 | -0.061 | -0.093 | -0.016 | -0.009 | 0.002  | -0.001 |
|   | Class 5 | 0.851 | 0.786 | 0.934 | 0.924 | -0.249 | -0.217 | -0.055 | -0.094 | 0.016  | 0.013  | -0.004 | -0.005 |
| 2 | Class 0 | 0.970 | 0.957 | 0.915 | 0.904 | -0.804 | -0.808 | -0.389 | -0.386 | -0.061 | -0.133 | -0.077 | -0.096 |
|   | Class 1 | 0.933 | 0.932 | 0.931 | 0.913 | 0.053  | -0.003 | -0.819 | -0.781 | -0.011 | -0.049 | -0.110 | -0.145 |
|   | Class 2 | 0.901 | 0.865 | 0.934 | 0.924 | -0.868 | -0.737 | -0.515 | -0.519 | -0.365 | -0.337 | -0.077 | -0.126 |
|   | Class 3 | 0.926 | 0.924 | 0.932 | 0.919 | -0.870 | -0.805 | -0.498 | -0.501 | -0.215 | -0.248 | -0.096 | -0.153 |
|   | Class 4 | 0.885 | 0.867 | 0.938 | 0.927 | -0.729 | -0.707 | -0.461 | -0.463 | -0.042 | -0.077 | -0.076 | -0.116 |
|   | Class 5 | 0.851 | 0.786 | 0.934 | 0.924 | -0.845 | -0.769 | -0.511 | -0.508 | -0.229 | -0.188 | -0.106 | -0.163 |
| 3 | Class 0 | 0.970 | 0.957 | 0.915 | 0.904 | -0.896 | -0.904 | -0.824 | -0.832 | -0.647 | -0.688 | -0.517 | -0.544 |
|   | Class 1 | 0.933 | 0.932 | 0.931 | 0.913 | -0.901 | -0.916 | -0.835 | -0.832 | -0.568 | -0.607 | -0.609 | -0.630 |
|   | Class 2 | 0.901 | 0.865 | 0.934 | 0.924 | -0.868 | -0.845 | -0.838 | -0.851 | -0.605 | -0.600 | -0.589 | -0.619 |
|   | Class 3 | 0.926 | 0.924 | 0.932 | 0.919 | -0.868 | -0.896 | -0.830 | -0.840 | -0.567 | -0.605 | -0.567 | -0.596 |
|   | Class 4 | 0.885 | 0.867 | 0.938 | 0.927 | -0.800 | -0.811 | -0.849 | -0.857 | -0.502 | -0.522 | -0.513 | -0.544 |
|   | Class 5 | 0.851 | 0.786 | 0.934 | 0.924 | 0.022  | 0.081  | -0.865 | -0.881 | -0.155 | -0.124 | -0.561 | -0.596 |
| 4 | Class 0 | 0.970 | 0.957 | 0.915 | 0.904 | -0.650 | -0.703 | -0.698 | -0.764 | -0.608 | -0.621 | -0.499 | -0.510 |
|   | Class 1 | 0.933 | 0.932 | 0.931 | 0.913 | -0.845 | -0.884 | -0.667 | -0.725 | -0.491 | -0.519 | -0.480 | -0.491 |
|   | Class 2 | 0.901 | 0.865 | 0.934 | 0.924 | -0.858 | -0.824 | -0.772 | -0.809 | -0.488 | -0.497 | -0.506 | -0.523 |
|   | Class 3 | 0.926 | 0.924 | 0.932 | 0.919 | -0.700 | -0.808 | -0.663 | -0.739 | -0.534 | -0.546 | -0.512 | -0.528 |
|   | Class 4 | 0.885 | 0.867 | 0.938 | 0.927 | -0.239 | -0.514 | -0.754 | -0.797 | -0.304 | -0.307 | -0.452 | -0.471 |
|   | Class 5 | 0.851 | 0.786 | 0.934 | 0.924 | -0.612 | -0.463 | -0.692 | -0.765 | -0.047 | -0.038 | -0.525 | -0.541 |
| 5 | Class 0 | 0.970 | 0.957 | 0.915 | 0.904 | -0.838 | -0.852 | -0.492 | -0.630 | -0.695 | -0.688 | -0.554 | -0.555 |
|   | Class 1 | 0.933 | 0.932 | 0.931 | 0.913 | -0.387 | -0.563 | -0.683 | -0.714 | -0.552 | -0.564 | -0.605 | -0.599 |
|   | Class 2 | 0.901 | 0.865 | 0.934 | 0.924 | -0.702 | -0.700 | -0.634 | -0.690 | -0.472 | -0.470 | -0.607 | -0.605 |
|   | Class 3 | 0.926 | 0.924 | 0.932 | 0.919 | -0.361 | -0.507 | -0.615 | -0.692 | -0.567 | -0.575 | -0.538 | -0.539 |
|   | Class 4 | 0.885 | 0.867 | 0.938 | 0.927 | -0.873 | -0.844 | -0.525 | -0.650 | -0.668 | -0.653 | -0.594 | -0.594 |
|   | Class 5 | 0.851 | 0.786 | 0.934 | 0.924 | -0.637 | -0.573 | -0.588 | -0.681 | -0.069 | -0.022 | -0.548 | -0.553 |
| 6 | Class 0 | 0.970 | 0.957 | 0.915 | 0.904 | -0.720 | -0.775 | -0.829 | -0.830 | -0.484 | -0.499 | -0.318 | -0.322 |
|   | Class 1 | 0.933 | 0.932 | 0.931 | 0.913 | -0.871 | -0.887 | -0.750 | -0.768 | -0.176 | -0.195 | -0.499 | -0.499 |
|   | Class 2 | 0.901 | 0.865 | 0.934 | 0.924 | -0.895 | -0.860 | -0.735 | -0.773 | -0.680 | -0.638 | -0.335 | -0.348 |
|   | Class 3 | 0.926 | 0.924 | 0.932 | 0.919 | -0.863 | -0.884 | -0.772 | -0.793 | -0.418 | -0.431 | -0.379 | -0.381 |
|   | Class 4 | 0.885 | 0.867 | 0.938 | 0.927 | -0.621 | -0.669 | -0.743 | -0.784 | -0.430 | -0.435 | -0.247 | -0.262 |
|   | Class 5 | 0.851 | 0.786 | 0.934 | 0.924 | -0.143 | -0.086 | -0.808 | -0.831 | -0.114 | -0.070 | -0.474 | -0.478 |

Table 11: Evaluation of layer selection on *GPT-2* model on the *Emotions* dataset using neuron and range masking techniques. 20% Neurons selected. Here, **Acc** represents class accuracy, **Conf** denotes class prediction probability, and **CAcc** and **CConf** refer to average accuracy and average class prediction probability across other classes, respectively. The *Base Values* indicate the baseline model performance, while *Activation Range Masking* and *Neuron Masking* show deviations from the baseline performance.

| Layer | Class | Base Values | | | | Neuron Masking | | | | Activation Range Masking | | | |
|---|---|---|---|---|---|---|---|---|---|---|---|---|---|
| | | Acc | Conf | CAcc | CConf | ΔAcc | ΔConf | ΔCAcc | ΔCConf | ΔAcc | ΔConf | ΔCAcc | ΔCConf |
| 7 | Class 0 | 0.970 | 0.957 | 0.915 | 0.904 | -0.908 | -0.901 | -0.752 | -0.753 | -0.527 | -0.538 | -0.492 | -0.498 |
| | Class 1 | 0.933 | 0.932 | 0.931 | 0.913 | -0.884 | -0.895 | -0.743 | -0.729 | -0.484 | -0.509 | -0.330 | -0.338 |
| | Class 2 | 0.901 | 0.865 | 0.934 | 0.924 | -0.866 | -0.835 | -0.767 | -0.765 | -0.451 | -0.442 | -0.336 | -0.355 |
| | Class 3 | 0.926 | 0.924 | 0.932 | 0.919 | -0.786 | -0.819 | -0.641 | -0.666 | -0.445 | -0.457 | -0.331 | -0.346 |
| | Class 4 | 0.885 | 0.867 | 0.938 | 0.927 | -0.626 | -0.618 | -0.810 | -0.817 | -0.341 | -0.335 | -0.521 | -0.532 |
| | Class 5 | 0.851 | 0.786 | 0.934 | 0.924 | 0.106 | 0.147 | -0.810 | -0.811 | 0.102 | 0.107 | -0.547 | -0.553 |
| 8 | Class 0 | 0.970 | 0.957 | 0.915 | 0.904 | -0.776 | -0.791 | -0.209 | -0.291 | -0.191 | -0.312 | -0.082 | -0.114 |
| | Class 1 | 0.933 | 0.932 | 0.931 | 0.913 | -0.585 | -0.667 | -0.412 | -0.441 | -0.591 | -0.644 | -0.199 | -0.227 |
| | Class 2 | 0.901 | 0.865 | 0.934 | 0.924 | -0.692 | -0.716 | -0.469 | -0.496 | -0.560 | -0.562 | -0.468 | -0.486 |
| | Class 3 | 0.926 | 0.924 | 0.932 | 0.919 | -0.657 | -0.714 | -0.415 | -0.464 | -0.468 | -0.503 | -0.230 | -0.266 |
| | Class 4 | 0.885 | 0.867 | 0.938 | 0.927 | -0.501 | -0.509 | -0.531 | -0.569 | -0.201 | -0.234 | -0.258 | -0.290 |
| | Class 5 | 0.851 | 0.786 | 0.934 | 0.924 | -0.092 | -0.050 | -0.634 | -0.647 | 0.065 | 0.058 | -0.279 | -0.308 |
| 9 | Class 0 | 0.970 | 0.957 | 0.915 | 0.904 | -0.759 | -0.768 | -0.311 | -0.351 | -0.610 | -0.661 | -0.307 | -0.328 |
| | Class 1 | 0.933 | 0.932 | 0.931 | 0.913 | -0.570 | -0.713 | -0.319 | -0.346 | -0.906 | -0.910 | -0.267 | -0.298 |
| | Class 2 | 0.901 | 0.865 | 0.934 | 0.924 | -0.424 | -0.520 | -0.504 | -0.531 | -0.635 | -0.643 | -0.579 | -0.595 |
| | Class 3 | 0.926 | 0.924 | 0.932 | 0.919 | -0.810 | -0.834 | -0.501 | -0.502 | -0.759 | -0.772 | -0.502 | -0.516 |
| | Class 4 | 0.885 | 0.867 | 0.938 | 0.927 | -0.358 | -0.357 | -0.476 | -0.481 | -0.587 | -0.566 | -0.519 | -0.527 |
| | Class 5 | 0.851 | 0.786 | 0.934 | 0.924 | -0.133 | -0.101 | -0.546 | -0.554 | 0.106 | 0.104 | -0.450 | -0.462 |
| 10 | Class 0 | 0.970 | 0.957 | 0.915 | 0.904 | -0.733 | -0.741 | -0.105 | -0.126 | -0.624 | -0.659 | -0.146 | -0.163 |
| | Class 1 | 0.933 | 0.932 | 0.931 | 0.913 | -0.389 | -0.671 | -0.178 | -0.209 | -0.899 | -0.911 | -0.254 | -0.285 |
| | Class 2 | 0.901 | 0.865 | 0.934 | 0.924 | -0.230 | -0.513 | -0.116 | -0.224 | -0.699 | -0.735 | -0.409 | -0.451 |
| | Class 3 | 0.926 | 0.924 | 0.932 | 0.919 | -0.434 | -0.687 | -0.081 | -0.133 | -0.898 | -0.905 | -0.401 | -0.455 |
| | Class 4 | 0.885 | 0.867 | 0.938 | 0.927 | -0.489 | -0.506 | -0.188 | -0.256 | -0.140 | -0.186 | -0.063 | -0.102 |
| | Class 5 | 0.851 | 0.786 | 0.934 | 0.924 | -0.306 | -0.243 | -0.157 | -0.240 | 0.063 | 0.010 | -0.095 | -0.127 |
| 11 | Class 0 | 0.970 | 0.957 | 0.915 | 0.904 | -0.358 | -0.496 | -0.382 | -0.414 | -0.301 | -0.441 | -0.121 | -0.148 |
| | Class 1 | 0.933 | 0.932 | 0.931 | 0.913 | -0.800 | -0.857 | -0.078 | -0.123 | -0.858 | -0.875 | -0.128 | -0.162 |
| | Class 2 | 0.901 | 0.865 | 0.934 | 0.924 | -0.897 | -0.861 | -0.416 | -0.450 | -0.901 | -0.864 | -0.464 | -0.500 |
| | Class 3 | 0.926 | 0.924 | 0.932 | 0.919 | -0.923 | -0.921 | -0.427 | -0.470 | -0.913 | -0.914 | -0.354 | -0.393 |
| | Class 4 | 0.885 | 0.867 | 0.938 | 0.927 | -0.152 | -0.212 | -0.039 | -0.075 | -0.210 | -0.239 | -0.181 | -0.204 |
| | Class 5 | 0.851 | 0.786 | 0.934 | 0.924 | 0.047 | -0.028 | -0.131 | -0.173 | 0.053 | 0.002 | -0.142 | -0.159 |
| 12 | Class 0 | 0.970 | 0.957 | 0.915 | 0.904 | -0.550 | -0.603 | -0.013 | -0.003 | -0.542 | -0.594 | 0.005 | 0.012 |
| | Class 1 | 0.933 | 0.932 | 0.931 | 0.913 | -0.526 | -0.545 | 0.001 | 0.012 | -0.521 | -0.538 | -0.005 | -0.004 |
| | Class 2 | 0.901 | 0.865 | 0.934 | 0.924 | -0.416 | -0.402 | 0.002 | 0.006 | -0.419 | -0.407 | 0.007 | 0.006 |
| | Class 3 | 0.926 | 0.924 | 0.932 | 0.919 | -0.561 | -0.576 | -0.007 | 0.003 | -0.561 | -0.572 | 0.000 | 0.005 |
| | Class 4 | 0.885 | 0.867 | 0.938 | 0.927 | -0.655 | -0.658 | -0.042 | -0.034 | -0.657 | -0.659 | -0.011 | -0.003 |
| | Class 5 | 0.851 | 0.786 | 0.934 | 0.924 | -0.718 | -0.672 | -0.300 | -0.297 | -0.718 | -0.672 | -0.267 | -0.266 |

Table 12: **Dampening intervention results** on Llama-3.2-3B (DBPedia-14): comparison of neuron and range masking. 30% neurons were selected. Dampening factor used is $a = 0.125$. **Acc** represents class accuracy, **Conf** denotes class prediction probability, and **CAcc** and **CConf** refer to average accuracy and average class prediction probability across other classes, respectively. The *Base Values* indicate the baseline model performance, while *Neuron Masking* and *Activation Range Masking* show deviations from the baseline performance. PPL Δ and MMLU Δ show changes in perplexity and MMLU scores, respectively.

| Class | Base Values | | | | Neuron Masking | | | | | | Activation Range Masking | | | | | |
|---|---|---|---|---|---|---|---|---|---|---|---|---|---|---|---|---|
| | Acc | Conf | CAcc | CConf | ΔAcc | ΔConf | ΔCAcc | ΔCConf | ΔPPL | ΔMMLU | ΔAcc | ΔConf | ΔCAcc | ΔCConf | ΔPPL | ΔMMLU |
| Class 0 | 1.000 | 0.576 | 1.000 | 0.563 | -0.919 | -0.545 | -0.281 | -0.309 | 3.161 | -0.020 | -0.924 | -0.545 | **-0.276** | **-0.285** | **0.640** | **-0.010** |
| Class 1 | 1.000 | 0.526 | 1.000 | 0.567 | -0.988 | -0.467 | -0.246 | -0.270 | 3.578 | -0.015 | -0.805 | -0.466 | **-0.193** | **-0.206** | **0.725** | **0.015** |
| Class 2 | 1.000 | 0.441 | 1.000 | 0.575 | -0.864 | -0.391 | -0.461 | -0.323 | 2.891 | -0.030 | -0.869 | -0.389 | **-0.346** | **-0.282** | **0.718** | **0.005** |
| Class 3 | 1.000 | 0.461 | 1.000 | 0.573 | -0.974 | -0.439 | -0.411 | -0.346 | 3.036 | -0.025 | -0.970 | -0.438 | **-0.282** | **-0.283** | **0.653** | **0.010** |
| Class 4 | 1.000 | 0.839 | 1.000 | 0.541 | -0.382 | -0.597 | -0.367 | -0.317 | 2.997 | 0.000 | -0.382 | -0.597 | **-0.334** | **-0.284** | **0.691** | **0.020** |
| Class 5 | 1.000 | 0.339 | 1.000 | 0.568 | -0.970 | -0.326 | -0.239 | -0.246 | 3.503 | 0.010 | -0.970 | -0.325 | **-0.197** | **-0.187** | **0.810** | **0.015** |
| Class 6 | 1.000 | 0.810 | 1.000 | 0.545 | -0.233 | -0.638 | -0.194 | -0.276 | 3.126 | -0.010 | -0.241 | -0.637 | **-0.174** | **-0.203** | **0.697** | **-0.010** |
| Class 7 | 1.000 | 0.595 | 1.000 | 0.562 | -0.210 | -0.382 | -0.206 | -0.226 | 3.037 | 0.000 | -0.179 | -0.376 | **-0.123** | **-0.143** | **0.546** | **0.015** |
| Class 8 | 1.000 | 0.417 | 1.000 | 0.574 | -0.310 | -0.416 | -0.335 | -0.297 | 3.001 | 0.020 | -0.346 | -0.416 | **-0.200** | **-0.187** | **0.624** | **0.015** |
| Class 9 | 1.000 | 0.526 | 1.000 | 0.567 | -0.820 | -0.465 | -0.327 | -0.264 | 3.369 | -0.030 | -0.809 | -0.463 | **-0.213** | **-0.189** | **0.596** | **0.000** |
| Class 10 | 1.000 | 0.505 | 1.000 | 0.569 | -0.691 | -0.466 | -0.389 | -0.314 | 3.732 | **0.000** | -0.696 | -0.465 | **-0.267** | **-0.198** | **0.695** | **-0.015** |
| Class 11 | 1.000 | 0.497 | 1.000 | 0.569 | -0.873 | -0.432 | -0.472 | -0.289 | 3.070 | -0.030 | -0.865 | -0.427 | **-0.335** | **-0.205** | **0.594** | **-0.015** |
| Class 12 | 1.000 | 0.573 | 1.000 | 0.563 | -0.720 | -0.452 | -0.295 | -0.221 | 3.410 | -0.045 | -0.723 | -0.451 | **-0.190** | **-0.163** | **0.595** | **0.035** |
| Class 13 | 1.000 | 0.567 | 1.000 | 0.564 | -0.951 | -0.537 | -0.226 | -0.189 | 2.995 | 0.000 | -0.955 | -0.536 | **-0.157** | **-0.150** | **0.672** | **0.005** |

Table 13: **Mean replacement intervention** results on Llama-3.2-3B (AG-News): comparison of neuron and range masking. 30% neurons were selected 30% neurons were selected. Mean Activation $\mu$ is used as a replacement value. **Acc** represents class accuracy, **Conf** denotes class prediction probability, and **CAcc** and **CConf** refer to average accuracy and average class prediction probability across other classes, respectively. The *Base Values* indicate the baseline model performance, while *Neuron Masking* and *Activation Range Masking* show deviations from the baseline performance. PPL $\Delta$ and MMLU $\Delta$ show changes in perplexity and MMLU scores, respectively.

| Class | Base Values | | | | Neuron Masking | | | | | | Activation Range Masking | | | | | |
|---|---|---|---|---|---|---|---|---|---|---|---|---|---|---|---|---|
| | Acc | Conf | CAcc | CConf | $\Delta$Acc | $\Delta$Conf | $\Delta$CAcc | $\Delta$CConf | $\Delta$PPL | $\Delta$MMLU | $\Delta$Acc | $\Delta$Conf | $\Delta$CAcc | $\Delta$CConf | $\Delta$PPL | $\Delta$MMLU |
| Class 0 | 1.000 | 0.718 | 1.000 | 0.448 | -1.000 | -0.716 | -0.271 | -0.414 | 20.748 | -0.005 | -0.979 | -0.715 | **-0.268** | **-0.334** | **0.474** | **0.020** |
| Class 1 | 1.000 | 0.474 | 1.000 | 0.533 | -0.932 | -0.460 | -0.404 | -0.508 | 22.865 | -0.005 | -0.894 | -0.454 | **-0.360** | **-0.427** | **0.445** | **-0.010** |
| Class 2 | 1.000 | 0.489 | 1.000 | 0.523 | -1.000 | -0.488 | -0.376 | -0.493 | 23.255 | 0.005 | -0.953 | -0.487 | **-0.340** | **-0.419** | **0.394** | **0.010** |
| Class 3 | 1.000 | 0.373 | 1.000 | 0.555 | -1.000 | -0.372 | -0.455 | -0.512 | 22.854 | 0.000 | -0.975 | -0.371 | **-0.350** | **-0.400** | **0.413** | **0.010** |

Table 14: **Mean replacement intervention** results on Llama-3.2-3B (Emotions): comparison of neuron and range masking. 30% neurons were selected 30% neurons were selected. Mean Activation $\mu$ is used as replacement value. **Acc** represents class accuracy, **Conf** denotes class prediction probability, and **CAcc** and **CConf** refer to average accuracy and average class prediction probability across other classes, respectively. The *Base Values* indicate the baseline model performance, while *Neuron Masking* and *Activation Range Masking* show deviations from the baseline performance. PPL $\Delta$ and MMLU $\Delta$ show changes in perplexity and MMLU scores, respectively.

| Class | Base Values | | | | Neuron Masking | | | | | | Activation Range Masking | | | | | |
|---|---|---|---|---|---|---|---|---|---|---|---|---|---|---|---|---|
| | Acc | Conf | CAcc | CConf | $\Delta$Acc | $\Delta$Conf | $\Delta$CAcc | $\Delta$CConf | $\Delta$PPL | $\Delta$MMLU | $\Delta$Acc | $\Delta$Conf | $\Delta$CAcc | $\Delta$CConf | $\Delta$PPL | $\Delta$MMLU |
| Class 0 | 1.000 | 0.531 | 1.000 | 0.546 | -0.902 | -0.511 | -0.801 | -0.503 | 7.205 | -0.010 | -0.905 | -0.510 | **-0.692** | **-0.487** | **0.494** | **-0.015** |
| Class 1 | 1.000 | 0.449 | 1.000 | 0.576 | -0.964 | -0.441 | -0.658 | -0.478 | 7.126 | 0.005 | -0.961 | -0.441 | **-0.677** | **-0.451** | **0.535** | **-0.025** |
| Class 2 | 1.000 | 0.479 | 1.000 | 0.545 | -0.972 | -0.477 | -0.689 | -0.447 | 7.079 | 0.015 | -0.972 | -0.476 | **-0.671** | **-0.427** | **0.506** | **0.025** |
| Class 3 | 1.000 | 0.638 | 1.000 | 0.510 | -0.680 | -0.563 | -0.703 | -0.441 | 7.275 | -0.005 | -0.702 | -0.563 | **-0.643** | **-0.423** | **0.507** | **0.010** |
| Class 4 | 1.000 | 0.627 | 1.000 | 0.533 | -0.890 | -0.603 | -0.550 | -0.412 | 6.903 | 0.010 | -0.593 | -0.603 | **-0.461** | **-0.354** | **0.462** | **-0.010** |
| Class 5 | 1.000 | 0.557 | 1.000 | 0.540 | -0.769 | -0.506 | -0.592 | -0.416 | 6.956 | 0.025 | -0.750 | -0.502 | **-0.552** | **-0.397** | **0.507** | **-0.005** |

Table 15: **Mean replacement intervention** results on Llama-3.2-3B (DBPedia-14): comparison of neuron and range masking. 30% neurons were selected 30% neurons were selected. Mean Activation $\mu$ is used as replacement value. **Acc** represents class accuracy, **Conf** denotes class prediction probability, and **CAcc** and **CConf** refer to average accuracy and average class prediction probability across other classes, respectively. The *Base Values* indicate the baseline model performance, while *Neuron Masking* and *Activation Range Masking* show deviations from the baseline performance. PPL $\Delta$ and MMLU $\Delta$ show changes in perplexity and MMLU scores, respectively.

| Class | Base Values | | | | Neuron Masking | | | | | | Activation Range Masking | | | | | |
|---|---|---|---|---|---|---|---|---|---|---|---|---|---|---|---|---|
| | Acc | Conf | CAcc | CConf | $\Delta$Acc | $\Delta$Conf | $\Delta$CAcc | $\Delta$CConf | $\Delta$PPL | $\Delta$MMLU | $\Delta$Acc | $\Delta$Conf | $\Delta$CAcc | $\Delta$CConf | $\Delta$PPL | $\Delta$MMLU |
| Class 0 | 1.000 | 0.576 | 1.000 | 0.563 | -1.000 | -0.576 | -0.685 | -0.554 | 7.681 | -0.025 | -1.000 | -0.576 | **-0.551** | **-0.545** | **0.687** | **-0.005** |
| Class 1 | 1.000 | 0.526 | 1.000 | 0.567 | -1.000 | -0.526 | -0.554 | -0.550 | 8.437 | -0.030 | -1.000 | -0.526 | **-0.356** | **-0.517** | **0.583** | **0.015** |
| Class 2 | 1.000 | 0.441 | 1.000 | 0.575 | -0.995 | -0.441 | -0.697 | -0.556 | 7.567 | -0.015 | -0.995 | -0.440 | **-0.574** | **-0.536** | **0.520** | **-0.010** |
| Class 3 | 1.000 | 0.461 | 1.000 | 0.573 | -1.000 | -0.461 | -0.766 | -0.561 | 8.005 | -0.015 | -1.000 | -0.461 | **-0.538** | **-0.534** | **0.543** | **0.010** |
| Class 4 | 1.000 | 0.839 | 1.000 | 0.541 | -1.000 | -0.838 | -0.724 | -0.528 | 8.239 | **0.010** | -0.995 | -0.838 | **-0.502** | **-0.503** | **0.565** | 0.005 |
| Class 5 | 1.000 | 0.339 | 1.000 | 0.568 | -1.000 | -0.339 | -0.616 | -0.551 | 7.753 | **0.010** | -1.000 | -0.339 | **-0.382** | **-0.510** | **0.552** | 0.005 |
| Class 6 | 1.000 | 0.810 | 1.000 | 0.545 | -0.313 | -0.805 | -0.549 | -0.531 | 7.880 | -0.005 | -0.292 | -0.805 | **-0.336** | **-0.499** | **0.547** | **0.020** |
| Class 7 | 1.000 | 0.595 | 1.000 | 0.562 | -1.000 | -0.592 | -0.491 | -0.535 | 7.413 | -0.010 | -0.995 | -0.591 | **-0.267** | **-0.449** | **0.462** | **0.000** |
| Class 8 | 1.000 | 0.417 | 1.000 | 0.574 | -0.928 | -0.414 | -0.632 | -0.556 | 7.688 | 0.015 | -0.934 | -0.414 | **-0.298** | **-0.489** | **0.495** | 0.015 |
| Class 9 | 1.000 | 0.526 | 1.000 | 0.567 | -1.000 | -0.526 | -0.611 | -0.544 | 8.057 | -0.035 | -1.000 | -0.526 | **-0.370** | **-0.482** | **0.467** | **0.015** |
| Class 10 | 1.000 | 0.505 | 1.000 | 0.569 | -0.998 | -0.505 | -0.642 | -0.558 | 8.791 | -0.020 | -0.998 | -0.505 | **-0.406** | **-0.485** | **0.484** | **0.005** |
| Class 11 | 1.000 | 0.497 | 1.000 | 0.569 | -1.000 | -0.497 | -0.719 | -0.543 | 7.903 | **0.025** | -1.000 | -0.497 | **-0.447** | **-0.459** | **0.397** | -0.005 |
| Class 12 | 1.000 | 0.573 | 1.000 | 0.563 | -0.904 | -0.572 | -0.629 | -0.543 | 8.046 | -0.005 | -0.896 | -0.571 | **-0.375** | **-0.484** | **0.425** | **0.000** |
| Class 13 | 1.000 | 0.567 | 1.000 | 0.564 | -1.000 | -0.566 | -0.526 | -0.533 | 7.543 | -0.025 | -0.998 | -0.566 | **-0.341** | **-0.481** | **0.464** | **-0.010** |

Table 16: Adaptive dampening intervention results on Llama-3.2-3B (DBPedia-14): comparison of neuron and range masking. 30% neurons were selected. Adaptive Dampening factor used is $\beta = 0.5$. **Acc** represents class accuracy, **Conf** denotes class prediction probability, and **CAcc** and **CConf** refer to average accuracy and average class prediction probability across other classes, respectively. The *Base Values* indicate the baseline model performance, while *Neuron Masking* and *Activation Range Masking* show deviations from the baseline performance. PPL $\Delta$ and MMLU $\Delta$ show changes in perplexity and MMLU scores, respectively.

| Class | Base Values | | | | Activation Range Masking | | | | | |
|---|---|---|---|---|---|---|---|---|---|---|
| | Acc | Conf | CAcc | CConf | $\Delta$Acc | $\Delta$Conf | $\Delta$CAcc | $\Delta$CConf | $\Delta$PPL | $\Delta$MMLU |
| Class 0 | 1.000 | 0.576 | 1.000 | 0.563 | -0.927 | -0.543 | -0.215 | -0.217 | 0.487 | -0.015 |
| Class 1 | 1.000 | 0.526 | 1.000 | 0.567 | -0.791 | -0.451 | -0.134 | -0.109 | 0.543 | 0.000 |
| Class 2 | 1.000 | 0.441 | 1.000 | 0.575 | -0.828 | -0.380 | -0.277 | -0.215 | 0.540 | -0.010 |
| Class 3 | 1.000 | 0.461 | 1.000 | 0.573 | -0.958 | -0.432 | -0.230 | -0.214 | 0.492 | 0.010 |
| Class 4 | 1.000 | 0.839 | 1.000 | 0.541 | -0.346 | -0.579 | -0.261 | -0.218 | 0.521 | 0.015 |
| Class 5 | 1.000 | 0.339 | 1.000 | 0.568 | -0.960 | -0.319 | -0.140 | -0.116 | 0.609 | -0.015 |
| Class 6 | 1.000 | 0.810 | 1.000 | 0.545 | -0.236 | -0.613 | -0.130 | -0.122 | 0.524 | -0.010 |
| Class 7 | 1.000 | 0.595 | 1.000 | 0.562 | -0.243 | -0.388 | -0.108 | -0.080 | 0.408 | 0.005 |
| Class 8 | 1.000 | 0.417 | 1.000 | 0.574 | -0.440 | -0.414 | -0.152 | -0.088 | 0.465 | 0.030 |
| Class 9 | 1.000 | 0.526 | 1.000 | 0.567 | -0.799 | -0.459 | -0.182 | -0.131 | 0.445 | 0.005 |
| Class 10 | 1.000 | 0.505 | 1.000 | 0.569 | -0.684 | -0.451 | -0.222 | -0.130 | 0.513 | -0.010 |
| Class 11 | 1.000 | 0.497 | 1.000 | 0.569 | -0.836 | -0.420 | -0.308 | -0.155 | 0.440 | -0.005 |
| Class 12 | 1.000 | 0.573 | 1.000 | 0.563 | -0.720 | -0.451 | -0.172 | -0.095 | 0.444 | 0.025 |
| Class 13 | 1.000 | 0.567 | 1.000 | 0.564 | -0.941 | -0.530 | -0.142 | -0.098 | 0.502 | 0.010 |

Table 17: Performance metrics for varying $\tau$ values.

| $\tau$ | Acc | Conf | CAcc | CConf |
|---|---|---|---|---|
| 0.3 | 0.9021 | 0.8858 | 0.9452 | 0.9358 |
| 0.6 | 0.8439 | 0.8185 | 0.9424 | 0.9327 |
| 0.9 | 0.7801 | 0.7486 | 0.9391 | 0.9263 |
| 1.2 | 0.7295 | 0.6950 | 0.9340 | 0.9174 |
| 1.5 | 0.6834 | 0.6482 | 0.9337 | 0.9093 |
| 1.8 | 0.6424 | 0.6141 | 0.9331 | 0.9000 |
| 2.1 | 0.6184 | 0.5926 | 0.9327 | 0.8910 |
| 2.4 | 0.6126 | 0.5858 | 0.9314 | 0.8846 |
| 2.7 | 0.6024 | 0.5798 | 0.9280 | 0.8800 |
| 3.0 | 0.5971 | 0.5776 | 0.9234 | 0.8777 |
| 3.3 | 0.5963 | 0.5786 | 0.9173 | 0.8753 |
| 3.6 | 0.5970 | 0.5794 | 0.9097 | 0.8729 |
| 3.9 | 0.5976 | 0.5802 | 0.9020 | 0.8698 |
| 4.2 | 0.5967 | 0.5798 | 0.8908 | 0.8642 |
| 4.5 | 0.5967 | 0.5798 | 0.8795 | 0.8577 |

Table 18: Evaluation of Llama on selected concepts for max and probeless rankings (top 20% neurons).

| Concept | Max | | | | | Probeless | | | | |
|---|---|---|---|---|---|---|---|---|---|---|
| | $\Delta$Acc | $\Delta$Conf | $\Delta$CAcc | $\Delta$CConf | $\Delta$PPL | $\Delta$Acc | $\Delta$Conf | $\Delta$CAcc | $\Delta$CConf | $\Delta$PPL |
| Pronouns | -0.79 | -0.51 | -0.35 | -0.17 | 8.94 | -0.73 | -0.45 | -0.24 | -0.13 | 2.03 |

Table 19: Evaluation of Llama on pronoun concepts using activation range masking (he, his, she, her, they, them)

| Concept | Neuron Masking | | | | | Activation Range Masking | | | | |
|---|---|---|---|---|---|---|---|---|---|---|
| | $\Delta$Acc | $\Delta$Conf | $\Delta$CAcc | $\Delta$CConf | $\Delta$PPL | $\Delta$Acc | $\Delta$Conf | $\Delta$CAcc | $\Delta$CConf | $\Delta$PPL |
| He | -0.94 | -0.59 | -0.10 | 0.06 | 2.67 | -0.95 | -0.60 | **-0.07** | **0.14** | **0.91** |
| She | -0.96 | -0.61 | -0.21 | -0.11 | 1.80 | -0.97 | -0.61 | **-0.15** | **0.03** | **0.68** |
| He/His | -0.80 | -0.50 | -0.12 | -0.00 | 1.52 | -0.80 | 0.46 | **0.00** | **0.17** | **0.32** |
| She/Her | -0.77 | -0.50 | -0.28 | -0.17 | 1.61 | -0.69 | 0.43 | **-0.03** | **0.05** | **0.36** |
| They | -0.50 | -0.27 | -0.40 | -0.30 | 2.80 | -0.61 | -0.31 | **-0.09** | **-0.03** | **0.67** |
| They/Them | -0.42 | -0.22 | -0.34 | -0.25 | 1.79 | -0.38 | -0.19 | **-0.04** | **0.03** | **0.42** |

Table 20: Evaluation of Llama on terminal punctuation concepts using activation range masking (?, !, .).

| Setting | Neuron Masking | | | | | Activation Range Masking | | | | |
|---|---|---|---|---|---|---|---|---|---|---|
| | $\Delta$Acc | $\Delta$Conf | $\Delta$CAcc | $\Delta$CConf | $\Delta$PPL | $\Delta$Acc | $\Delta$Conf | $\Delta$CAcc | $\Delta$CConf | $\Delta$PPL |
| ? vs !, . | -0.94 | -0.58 | -0.23 | -0.13 | 1.72 | -0.89 | -0.54 | **-0.09** | **-0.04** | **0.54** |
| ! vs ?, . | -0.38 | -0.36 | -0.59 | -0.32 | 1.70 | -0.42 | -0.33 | **-0.18** | **-0.08** | **0.27** |
| . vs !, ? | -0.83 | -0.45 | -0.55 | -0.38 | 1.04 | -0.66 | -0.37 | **-0.14** | **-0.04** | **0.21** |

Table 21: Evaluation of Llama on delimiter (brackets) concepts using activation range masking "( [ {"

| Setting | Neuron Masking | | | | | Activation Range Masking | | | | |
|---|---|---|---|---|---|---|---|---|---|---|
| | $\Delta$Acc | $\Delta$Conf | $\Delta$CAcc | $\Delta$CConf | $\Delta$PPL | $\Delta$Acc | $\Delta$Conf | $\Delta$CAcc | $\Delta$CConf | $\Delta$PPL |
| [ vs {, ( | -0.75 | -0.43 | -0.18 | -0.26 | 1.53 | -0.72 | -0.40 | **-0.10** | **-0.14** | **0.47** |
| { vs [, ( | -0.90 | -0.62 | -0.13 | -0.14 | 1.69 | -0.83 | -0.60 | **0.01** | **0.02** | **0.28** |
| ( vs [, { | -0.44 | -0.41 | -0.41 | -0.37 | 1.86 | -0.50 | -0.42 | **-0.16** | **-0.11** | **0.48** |

Table 22: Evaluation of Llama on subject-verb agreement concepts using activation range masking (do, does, has, have)

| Setting | Neuron Masking | | | | | Activation Range Masking | | | | |
|---|---|---|---|---|---|---|---|---|---|---|
| | $\Delta$Acc | $\Delta$Conf | $\Delta$CAcc | $\Delta$CConf | $\Delta$PPL | $\Delta$Acc | $\Delta$Conf | $\Delta$CAcc | $\Delta$CConf | $\Delta$PPL |
| do | -0.46 | -0.28 | -0.25 | -0.07 | 1.41 | -0.55 | -0.28 | **-0.08** | **-0.03** | **0.33** |
| does | -0.95 | -0.41 | -0.17 | -0.07 | 1.48 | -0.92 | -0.38 | **0.15** | **0.01** | **0.55** |
| has | -0.33 | -0.20 | -0.36 | -0.18 | 1.29 | -0.45 | -0.22 | **-0.04** | **-0.02** | **0.21** |
| have | -0.90 | -0.47 | -0.29 | -0.08 | 1.14 | -0.86 | -0.44 | **-0.11** | **0.08** | **0.26** |

Table 23: Evaluation of selected models on the *AG-News* dataset using neuron and range masking techniques. **Acc** represents class accuracy, **Conf** denotes class prediction probability, and **CAcc** and **CConf** refer to average accuracy and average class prediction probability across other classes, respectively. The *Base Values* indicate the baseline model performance, while *Activation Range Masking* and *Neuron Masking* show deviations from the baseline performance. For *GPT-2* 50% and for *Llama-3.2-3B* 30% neurons selected.

| Model | Class | Base Values | | | | Neuron Masking | | | | Activation Range Masking | | | |
|---|---|---|---|---|---|---|---|---|---|---|---|---|---|
| | | Acc | Conf | CAcc | CConf | ΔAcc | ΔConf | ΔCAcc | ΔCConf | ΔAcc | ΔConf | ΔCAcc | ΔCConf |
| BERT | Class 0 | 0.945 | 0.936 | 0.949 | 0.927 | -0.205 | -0.587 | **0.004** | -0.076 | -0.198 | -0.589 | 0.007 | **-0.010** |
| | Class 1 | 0.993 | 0.988 | 0.933 | 0.910 | -0.225 | -0.659 | 0.004 | -0.077 | -0.194 | -0.650 | **0.003** | **-0.012** |
| | Class 2 | 0.905 | 0.881 | 0.962 | 0.945 | -0.300 | -0.536 | 0.014 | -0.079 | -0.298 | -0.542 | 0.014 | **-0.009** |
| | Class 3 | 0.949 | 0.913 | 0.948 | 0.935 | -0.354 | -0.577 | 0.026 | -0.065 | -0.353 | -0.579 | **0.025** | **-0.005** |
| GPT-2 | Class 0 | 0.955 | 0.951 | 0.941 | 0.928 | -0.920 | -0.926 | -0.231 | -0.224 | -0.919 | -0.925 | **-0.019** | **-0.008** |
| | Class 1 | 0.986 | 0.981 | 0.931 | 0.917 | -0.926 | -0.931 | -0.253 | -0.257 | -0.912 | -0.916 | **-0.054** | **-0.069** |
| | Class 2 | 0.897 | 0.886 | 0.960 | 0.949 | -0.696 | -0.737 | -0.110 | **-0.132** | -0.678 | -0.725 | **-0.097** | -0.306 |
| | Class 3 | 0.940 | 0.916 | 0.946 | 0.939 | -0.940 | -0.916 | -0.024 | **-0.037** | -0.887 | -0.882 | **-0.080** | -0.510 |
| Llama-3.2-3B | Class 0 | 1.000 | 0.936 | 1.000 | 0.680 | -0.995 | -0.934 | -0.530 | -0.427 | -0.995 | -0.934 | **-0.345** | **-0.306** |
| | Class 1 | 1.000 | 0.742 | 1.000 | 0.744 | -0.870 | -0.680 | -0.615 | -0.599 | -0.875 | -0.681 | **-0.515** | **-0.503** |
| | Class 2 | 1.000 | 0.655 | 1.000 | 0.773 | -0.895 | -0.646 | -0.795 | -0.634 | -0.895 | -0.646 | **-0.655** | **-0.549** |
| | Class 3 | 1.000 | 0.642 | 1.000 | 0.778 | -0.975 | -0.641 | -0.698 | -0.630 | -0.975 | -0.640 | **-0.420** | **-0.459** |

Table 24: Evaluation of selected models on the *IMDB* dataset using neuron and range masking techniques. Here, **Acc** represents class accuracy, **Conf** denotes class prediction probability, and **CAcc** and **CConf** refer to average accuracy and average class prediction probability across other classes, respectively. The *Base Values* indicate the baseline model performance, while *Activation Range Masking* and *Neuron Masking* show deviations from the baseline performance.

| Model | Class | Base Values | | | | Neuron Masking | | | | Activation Range Masking | | | |
|---|---|---|---|---|---|---|---|---|---|---|---|---|---|
| | | Acc | Conf | CAcc | CConf | ΔAcc | ΔConf | ΔCAcc | ΔCConf | ΔAcc | ΔConf | ΔCAcc | ΔCConf |
| BERT | Class 0 | 0.930 | 0.908 | 0.926 | 0.901 | -0.169 | -0.352 | 0.061 | -0.066 | -0.163 | -0.359 | 0.059 | 0.035 |
| | Class 1 | 0.926 | 0.901 | 0.930 | 0.908 | -0.211 | -0.355 | 0.057 | -0.091 | -0.206 | -0.361 | 0.056 | 0.025 |
| GPT-2 | Class 0 | 0.965 | 0.941 | 0.940 | 0.922 | -0.935 | -0.922 | 0.050 | 0.057 | -0.905 | -0.901 | 0.055 | 0.046 |
| | Class 1 | 0.940 | 0.922 | 0.965 | 0.941 | -0.620 | -0.667 | 0.005 | 0.018 | -0.610 | -0.657 | 0.015 | 0.027 |
| Llama-3.2-3B | Class 0 | 1.000 | 0.619 | 1.000 | 0.500 | -0.643 | -0.448 | -0.515 | -0.287 | -0.640 | -0.446 | -0.502 | -0.278 |
| | Class 1 | 1.000 | 0.500 | 1.000 | 0.619 | -0.877 | -0.410 | -0.273 | -0.304 | -0.873 | -0.409 | -0.265 | -0.303 |

Table 25: Evaluation of selected models on the *SST2* dataset using neuron and range masking techniques. Here, **Acc** represents class accuracy, **Conf** denotes class prediction probability, and **CAcc** and **CConf** refer to average accuracy and average class prediction probability across other classes, respectively. The *Base Values* indicate the baseline model performance, while *Activation Range Masking* and *Neuron Masking* show deviations from the baseline performance.

| Model | Class | Base Values | | | | Neuron Masking | | | | Activation Range Masking | | | |
|---|---|---|---|---|---|---|---|---|---|---|---|---|---|
| | | Acc | Conf | CAcc | CConf | ΔAcc | ΔConf | ΔCAcc | ΔCConf | ΔAcc | ΔConf | ΔCAcc | ΔCConf |
| BERT | Class 0 | 0.890 | 0.882 | 0.930 | 0.925 | -0.058 | -0.308 | 0.029 | -0.047 | -0.075 | -0.329 | 0.031 | 0.036 |
| | Class 1 | 0.930 | 0.925 | 0.890 | 0.882 | -0.043 | -0.318 | 0.033 | -0.045 | -0.045 | -0.330 | 0.030 | 0.050 |
| GPT-2 | Class 0 | 0.950 | 0.937 | 0.981 | 0.978 | -0.142 | -0.158 | 0.010 | 0.012 | -0.142 | -0.167 | 0.009 | 0.010 |
| | Class 1 | 0.981 | 0.978 | 0.950 | 0.937 | -0.187 | -0.223 | 0.041 | 0.053 | -0.176 | -0.216 | 0.041 | 0.046 |
| Llama-3.2-3B | Class 0 | 1.000 | 0.620 | 1.000 | 0.690 | -0.532 | -0.459 | -0.420 | -0.424 | -0.532 | -0.456 | -0.404 | -0.415 |
| | Class 1 | 1.000 | 0.690 | 1.000 | 0.620 | -0.289 | -0.379 | -0.326 | -0.315 | -0.284 | -0.376 | -0.306 | -0.301 |

Table 26: Evaluation of selected models on the *Emotions* dataset using neuron and range masking techniques. Here, **Acc** represents class accuracy, **Conf** denotes class prediction probability, and **CAcc** and **CConf** refer to average accuracy and average class prediction probability across other classes, respectively. The *Base Values* indicate the baseline model performance, while *Activation Range Masking* and *Neuron Masking* show deviations from the baseline performance.

| Model | Class | Base Values | | | | Neuron Masking | | | | Activation Range Masking | | | |
|---|---|---|---|---|---|---|---|---|---|---|---|---|---|
| | | Acc | Conf | CAcc | CConf | $\Delta$Acc | $\Delta$Conf | $\Delta$CAcc | $\Delta$CConf | $\Delta$Acc | $\Delta$Conf | $\Delta$CAcc | $\Delta$CConf |
| BERT | Class 0 | 0.960 | 0.935 | 0.901 | 0.851 | -0.241 | -0.718 | 0.013 | -0.266 | -0.222 | -0.718 | 0.012 | -0.055 |
| | Class 1 | 0.942 | 0.904 | 0.905 | 0.861 | -0.223 | -0.691 | 0.028 | -0.254 | -0.213 | -0.692 | 0.032 | -0.064 |
| | Class 2 | 0.824 | 0.723 | 0.926 | 0.889 | -0.371 | -0.533 | 0.016 | -0.284 | -0.352 | -0.534 | 0.018 | -0.115 |
| | Class 3 | 0.927 | 0.873 | 0.916 | 0.876 | -0.247 | -0.664 | 0.010 | -0.256 | -0.240 | -0.667 | 0.012 | -0.057 |
| | Class 4 | 0.884 | 0.837 | 0.922 | 0.880 | -0.406 | -0.646 | 0.012 | -0.251 | -0.402 | -0.648 | 0.012 | -0.066 |
| | Class 5 | 0.591 | 0.566 | 0.929 | 0.886 | -0.303 | -0.392 | 0.004 | -0.299 | -0.303 | -0.397 | 0.005 | -0.090 |
| GPT-2 | Class 0 | 0.969 | 0.956 | 0.913 | 0.903 | -0.695 | -0.751 | -0.125 | -0.124 | -0.698 | -0.749 | -0.009 | -0.009 |
| | Class 1 | 0.939 | 0.938 | 0.925 | 0.908 | -0.879 | -0.882 | -0.019 | -0.009 | -0.879 | -0.880 | -0.016 | -0.015 |
| | Class 2 | 0.902 | 0.872 | 0.932 | 0.923 | -0.776 | -0.736 | -0.029 | -0.032 | -0.780 | -0.739 | -0.023 | -0.028 |
| | Class 3 | 0.910 | 0.905 | 0.932 | 0.921 | -0.713 | -0.714 | -0.006 | -0.007 | -0.715 | -0.716 | -0.002 | -0.001 |
| | Class 4 | 0.869 | 0.854 | 0.938 | 0.927 | -0.754 | -0.753 | -0.240 | -0.248 | -0.754 | -0.753 | -0.127 | -0.133 |
| | Class 5 | 0.857 | 0.798 | 0.932 | 0.923 | -0.587 | -0.601 | -0.301 | -0.308 | -0.587 | -0.601 | -0.280 | -0.289 |
| Llama-3.2-3B | Class 0 | 0.950 | 0.550 | 0.782 | 0.455 | -0.950 | -0.547 | -0.655 | -0.408 | -0.945 | -0.547 | -0.571 | -0.378 |
| | Class 1 | 0.905 | 0.498 | 0.804 | 0.473 | -0.855 | -0.495 | -0.743 | -0.433 | -0.867 | -0.494 | -0.607 | -0.404 |
| | Class 2 | 0.785 | 0.421 | 0.827 | 0.483 | -0.785 | -0.420 | -0.771 | -0.454 | -0.785 | -0.420 | -0.658 | -0.436 |
| | Class 3 | 0.790 | 0.482 | 0.833 | 0.476 | -0.760 | -0.477 | -0.635 | -0.423 | -0.755 | -0.476 | -0.544 | -0.402 |
| | Class 4 | 0.780 | 0.487 | 0.829 | 0.476 | -0.780 | -0.486 | -0.534 | -0.365 | -0.780 | -0.486 | -0.444 | -0.324 |
| | Class 5 | 0.536 | 0.296 | 0.855 | 0.498 | -0.417 | -0.284 | -0.751 | -0.465 | -0.429 | -0.282 | -0.653 | -0.434 |

Table 27: Evaluation of selected models on the *DBPedia-14* dataset using neuron and range masking techniques. Here, **Acc** represents class accuracy, **Conf** denotes class prediction probability, and **CAcc** and **CConf** refer to average accuracy and average class prediction probability across other classes, respectively. The *Base Values* indicate the baseline model performance, while *Activation Range Masking* and *Neuron Masking* show deviations from the baseline performance.

| Model | Class | Base Values | | | | Neuron Masking | | | | Activation Range Masking | | | |
|---|---|---|---|---|---|---|---|---|---|---|---|---|---|
| | | Acc | Conf | CAcc | CConf | $\Delta$Acc | $\Delta$Conf | $\Delta$CAcc | $\Delta$CConf | $\Delta$Acc | $\Delta$Conf | $\Delta$CAcc | $\Delta$CConf |
| BERT | Class 0 | 0.972 | 0.966 | 0.992 | 0.991 | -0.082 | -0.702 | 0.001 | -0.014 | -0.076 | -0.698 | 0.001 | -0.000 |
| | Class 1 | 0.987 | 0.986 | 0.991 | 0.990 | -0.030 | -0.778 | 0.000 | -0.017 | -0.018 | -0.770 | 0.000 | -0.000 |
| | Class 2 | 0.987 | 0.985 | 0.991 | 0.990 | -0.239 | -0.814 | 0.001 | -0.018 | -0.217 | -0.806 | 0.001 | -0.000 |
| | Class 3 | 0.997 | 0.997 | 0.990 | 0.989 | -0.008 | -0.766 | 0.000 | -0.019 | -0.001 | -0.731 | 0.000 | -0.000 |
| | Class 4 | 0.984 | 0.983 | 0.991 | 0.990 | -0.058 | -0.777 | 0.001 | -0.018 | -0.032 | -0.761 | 0.000 | -0.000 |
| | Class 5 | 0.995 | 0.995 | 0.990 | 0.989 | -0.007 | -0.795 | 0.000 | -0.017 | -0.001 | -0.771 | 0.000 | -0.000 |
| | Class 6 | 0.975 | 0.974 | 0.992 | 0.991 | -0.121 | -0.807 | 0.000 | -0.015 | -0.112 | -0.803 | 0.000 | -0.001 |
| | Class 7 | 0.994 | 0.994 | 0.990 | 0.989 | -0.028 | -0.789 | 0.000 | -0.017 | -0.010 | -0.767 | 0.000 | -0.000 |
| | Class 8 | 1.000 | 1.000 | 0.990 | 0.989 | -0.001 | -0.808 | 0.000 | -0.022 | 0.000 | -0.772 | 0.000 | -0.000 |
| | Class 9 | 0.999 | 0.998 | 0.990 | 0.989 | -0.004 | -0.837 | 0.000 | -0.019 | -0.001 | -0.811 | 0.000 | -0.000 |
| | Class 10 | 0.994 | 0.993 | 0.990 | 0.989 | -0.025 | -0.846 | 0.000 | -0.016 | -0.005 | -0.831 | 0.000 | -0.000 |
| | Class 11 | 0.997 | 0.997 | 0.990 | 0.989 | -0.013 | -0.751 | 0.000 | -0.017 | -0.001 | -0.726 | 0.000 | -0.000 |
| | Class 12 | 0.990 | 0.990 | 0.990 | 0.989 | -0.018 | -0.772 | 0.000 | -0.017 | -0.005 | -0.755 | 0.000 | -0.000 |
| | Class 13 | 0.994 | 0.994 | 0.990 | 0.989 | -0.009 | -0.740 | 0.001 | -0.017 | -0.001 | -0.721 | 0.000 | -0.000 |
| GPT-2 | Class 0 | 0.985 | 0.977 | 0.990 | 0.989 | -0.860 | -0.877 | -0.133 | -0.136 | -0.850 | -0.869 | -0.002 | -0.017 |
| | Class 1 | 0.995 | 0.992 | 0.990 | 0.988 | -0.500 | -0.567 | -0.180 | -0.192 | -0.460 | -0.544 | -0.023 | -0.024 |
| | Class 2 | 0.985 | 0.980 | 0.990 | 0.989 | -0.890 | -0.904 | -0.189 | -0.213 | -0.880 | -0.902 | -0.004 | -0.010 |
| | Class 3 | 0.995 | 0.995 | 0.990 | 0.987 | -0.900 | -0.933 | -0.145 | -0.143 | -0.900 | -0.927 | -0.008 | -0.017 |
| | Class 4 | 0.970 | 0.969 | 0.992 | 0.989 | -0.715 | -0.773 | -0.224 | -0.260 | -0.695 | -0.750 | -0.042 | -0.062 |
| | Class 5 | 0.995 | 0.993 | 0.990 | 0.988 | -0.315 | -0.446 | -0.127 | -0.192 | -0.290 | -0.432 | -0.013 | -0.025 |
| | Class 6 | 0.965 | 0.964 | 0.992 | 0.990 | -0.925 | -0.932 | -0.052 | -0.062 | -0.910 | -0.928 | -0.006 | -0.007 |
| | Class 7 | 1.000 | 0.998 | 0.989 | 0.987 | -0.815 | -0.865 | -0.003 | -0.008 | -0.775 | -0.846 | -0.026 | -0.057 |
| | Class 8 | 1.000 | 1.000 | 0.989 | 0.987 | -0.995 | -0.990 | -0.148 | -0.188 | -0.900 | -0.932 | -0.026 | -0.055 |
| | Class 9 | 1.000 | 1.000 | 0.989 | 0.987 | -0.975 | -0.979 | -0.250 | -0.268 | -0.955 | -0.958 | -0.020 | -0.049 |
| | Class 10 | 0.995 | 0.993 | 0.990 | 0.988 | -0.595 | -0.685 | -0.045 | -0.053 | -0.590 | -0.675 | -0.005 | -0.011 |
| | Class 11 | 0.985 | 0.984 | 0.990 | 0.988 | -0.210 | -0.453 | -0.094 | -0.118 | -0.135 | -0.396 | -0.015 | -0.034 |
| | Class 12 | 0.990 | 0.988 | 0.990 | 0.988 | -0.930 | -0.938 | -0.293 | -0.309 | -0.855 | -0.880 | -0.013 | -0.029 |
| | Class 13 | 1.000 | 0.999 | 0.989 | 0.987 | -0.985 | -0.986 | -0.393 | -0.416 | -0.945 | -0.981 | -0.018 | -0.044 |
| Llama-3.2-3B | Class 0 | 1.000 | 0.586 | 1.000 | 0.559 | -0.990 | -0.584 | -0.949 | -0.473 | -0.990 | -0.584 | -0.823 | -0.441 |
| | Class 1 | 1.000 | 0.533 | 1.000 | 0.563 | -1.000 | -0.528 | -0.870 | -0.446 | -0.970 | -0.528 | -0.706 | -0.371 |
| | Class 2 | 1.000 | 0.467 | 1.000 | 0.568 | -0.995 | -0.462 | -0.963 | -0.477 | -0.995 | -0.461 | -0.838 | -0.432 |
| | Class 3 | 1.000 | 0.460 | 1.000 | 0.569 | -0.995 | -0.459 | -0.981 | -0.486 | -0.995 | -0.459 | -0.815 | -0.420 |
| | Class 4 | 1.000 | 0.828 | 1.000 | 0.539 | -0.965 | -0.809 | -0.981 | -0.454 | -0.955 | -0.808 | -0.852 | -0.412 |
| | Class 5 | 1.000 | 0.349 | 1.000 | 0.568 | -1.000 | -0.348 | -0.882 | -0.429 | -0.989 | -0.347 | -0.585 | -0.346 |
| | Class 6 | 1.000 | 0.809 | 1.000 | 0.541 | -1.000 | -0.787 | -0.972 | -0.449 | -1.000 | -0.787 | -0.736 | -0.366 |
| | Class 7 | 1.000 | 0.599 | 1.000 | 0.558 | -0.855 | -0.588 | -0.918 | -0.410 | -0.860 | -0.586 | -0.489 | -0.274 |
| | Class 8 | 1.000 | 0.420 | 1.000 | 0.572 | -1.000 | -0.420 | -0.957 | -0.467 | -1.000 | -0.420 | -0.660 | -0.335 |
| | Class 9 | 1.000 | 0.527 | 1.000 | 0.563 | -1.000 | -0.524 | -0.842 | -0.435 | -0.995 | -0.523 | -0.552 | -0.320 |
| | Class 10 | 1.000 | 0.505 | 1.000 | 0.565 | -0.995 | -0.503 | -0.907 | -0.464 | -1.000 | -0.503 | -0.589 | -0.322 |
| | Class 11 | 1.000 | 0.505 | 1.000 | 0.565 | -0.975 | -0.501 | -0.862 | -0.416 | -0.970 | -0.501 | -0.579 | -0.313 |
| | Class 12 | 1.000 | 0.560 | 1.000 | 0.561 | -0.980 | -0.545 | -0.812 | -0.417 | -0.975 | -0.544 | -0.496 | -0.310 |
| | Class 13 | 1.000 | 0.587 | 1.000 | 0.559 | -0.990 | -0.584 | -0.722 | -0.406 | -0.985 | -0.584 | -0.588 | -0.337 |

Table 28: Comparison of Activation Steering(using mean diff:Average representation) and Activation range masking on Llama-3.2-3B (AG News). **Acc** represents class accuracy, **Conf** denotes class prediction probability, and **CAcc** and **CConf** refer to average accuracy and average class prediction probability across other classes, respectively. The *Base Values* indicate the baseline model performance, while *Neuron Masking* and *Activation Range Masking* show deviations from the baseline performance. PPL $\Delta$ and MMLU $\Delta$ show changes in perplexity and MMLU scores, respectively.

| Class | Base Values | | | | Activation Steering | | | | | | Activation Range Masking | | | | | |
|---|---|---|---|---|---|---|---|---|---|---|---|---|---|---|---|---|
| | Acc | Conf | CAcc | CConf | $\Delta$Acc | $\Delta$Conf | $\Delta$CAcc | $\Delta$CConf | $\Delta$PPL | $\Delta$MMLU | $\Delta$Acc | $\Delta$Conf | $\Delta$CAcc | $\Delta$CConf | $\Delta$PPL | $\Delta$MMLU |
| Class 0 | 1.000 | 0.712 | 1.000 | 0.442 | -0.033 | -0.199 | -0.156 | -0.093 | 1.253 | 0.050 | -0.950 | -0.695 | -0.188 | -0.195 | 0.593 | 0.015 |
| Class 1 | 1.000 | 0.472 | 1.000 | 0.529 | -0.983 | -0.390 | -0.011 | 0.041 | 0.955 | 0.005 | -0.975 | -0.465 | -0.338 | -0.265 | 0.703 | -0.010 |
| Class 2 | 1.000 | 0.488 | 1.000 | 0.522 | -1.000 | -0.418 | -0.350 | -0.014 | 0.880 | 0.000 | -0.912 | -0.476 | -0.287 | -0.247 | 0.573 | -0.015 |
| Class 3 | 1.000 | 0.384 | 1.000 | 0.561 | -0.983 | -0.320 | -0.033 | -0.015 | 1.364 | 0.015 | -0.887 | -0.367 | -0.392 | -0.360 | 0.684 | 0.005 |

Table 29: Comparison of Activation Steering(Using Last token representation) and Activation Range (Activation Steering) on Llama-3.2-3B (AG News). **Acc** represents class accuracy, **Conf** denotes class prediction probability, and **CAcc** and **CConf** refer to average accuracy and average class prediction probability across other classes, respectively. The *Base Values* indicate the baseline model performance, while *Activation Steering* and *Activation Range (Activation Steering)* show deviations from the baseline performance. PPL Δ and MMLU Δ show changes in perplexity and MMLU scores, respectively.

| Class | Base Values | | | | Activation Steering | | | | | | Activation Range (Activation Steering) | | | | | |
|---|---|---|---|---|---|---|---|---|---|---|---|---|---|---|---|---|
| | Acc | Conf | CAcc | CConf | ΔAcc | ΔConf | ΔCAcc | ΔCConf | ΔPPL | ΔMMLU | ΔAcc | ΔConf | ΔCAcc | ΔCConf | ΔPPL | ΔMMLU |
| Class 0 | 1.000 | 0.716 | 1.000 | 0.444 | -0.938 | -0.639 | -0.025 | 0.262 | 3.817 | -0.020 | -0.900 | -0.628 | -0.008 | 0.180 | 1.225 | -0.005 |
| Class 1 | 1.000 | 0.473 | 1.000 | 0.521 | -1.000 | -0.473 | -0.275 | 0.016 | 5.721 | 0.000 | -1.000 | -0.473 | -0.137 | 0.008 | 1.825 | 0.005 |
| Class 2 | 1.000 | 0.483 | 1.000 | 0.523 | -1.000 | -0.483 | -0.233 | -0.034 | 2.196 | -0.020 | -1.000 | -0.483 | -0.329 | -0.076 | 0.921 | 0.035 |
| Class 3 | 1.000 | 0.387 | 1.000 | 0.563 | -1.000 | -0.387 | -0.442 | -0.159 | 4.665 | 0.000 | -1.000 | -0.387 | -0.562 | -0.194 | 1.324 | 0.015 |

Table 30: comparison of Directional Ablation and Activation range masking on Llama-3.2-3B (AG News). **Acc** represents class accuracy, **Conf** denotes class prediction probability, and **CAcc** and **CConf** refer to average accuracy and average class prediction probability across other classes, respectively. The *Base Values* indicate the baseline model performance, while *Neuron Masking* and *Activation Range Masking* show deviations from the baseline performance. PPL Δ and MMLU Δ show changes in perplexity and MMLU scores, respectively.

| Class | Base Values | | | | Directional Ablation | | | | | | Activation Range Masking | | | | | |
|---|---|---|---|---|---|---|---|---|---|---|---|---|---|---|---|---|
| | Acc | Conf | CAcc | CConf | ΔAcc | ΔConf | ΔCAcc | ΔCConf | ΔPPL | ΔMMLU | ΔAcc | ΔConf | ΔCAcc | ΔCConf | ΔPPL | ΔMMLU |
| Class 0 | 1.000 | 0.716 | 1.000 | 0.444 | -1.000 | -0.716 | -0.917 | -0.442 | 982.484 | -0.210 | -0.950 | -0.699 | -0.188 | -0.197 | 0.593 | 0.015 |
| Class 1 | 1.000 | 0.473 | 1.000 | 0.521 | -1.000 | -0.473 | -1.000 | -0.521 | 1068.310 | -0.205 | -0.975 | -0.466 | -0.338 | -0.257 | 0.703 | -0.010 |
| Class 2 | 1.000 | 0.483 | 1.000 | 0.523 | -1.000 | -0.483 | -0.996 | -0.522 | 1025.228 | -0.205 | -0.912 | -0.472 | -0.287 | -0.248 | 0.573 | -0.015 |
| Class 3 | 1.000 | 0.387 | 1.000 | 0.563 | -1.000 | -0.387 | -0.979 | -0.562 | 901.877 | -0.225 | -0.887 | -0.371 | -0.392 | -0.362 | 0.684 | 0.005 |

Table 31: SAE feature selection statistics for DBPedia.

| SAE | Avg. Activating (Per Sample) | Unique Activating (Per Concept) | Consistent (Per Concept) | Features Ablated (Per Concept) |
|---|---|---|---|---|
| 16k / l0_285 | 226 | 1024 | 409 | 81 / 16K |
| 65k / l0_197 | 153 | 1318 | 296 | 59 / 65K |

Table 32: Comparison of full feature masking and activation-range masking applied to SAE features on DBPedia.

| Method | SAE | ΔAcc | ΔConf | ΔCAcc | ΔCConf | ΔPPL |
|---|---|---|---|---|---|---|
| Feature Masking | 16k / l0_285 | -0.323 | -0.153 | -0.169 | -0.129 | 9.510 |
| Activation Range Masking | 16k / l0_285 | -0.322 | -0.153 | **-0.140** | **-0.095** | **0.958** |
| Feature Masking | 65k / l0_197 | -0.392 | -0.236 | -0.175 | -0.216 | 6.203 |
| Activation Range Masking | 65k / l0_197 | -0.391 | -0.235 | **-0.098** | **-0.129** | **0.335** |

| Dataset | Sample Text | Label |
|---|---|---|
| **IMDB** | Before Dogma 95: when Lars used movies as art, not just a story. A beautiful painting about love and death. This is one of my favorite movies of all time. The color... The music... Just perfect. | Positive |
| **SST2** | badly-rendered cgi effects. | Negative |
| **AG News** | Wall St. Bears Claw Back Into the Black (Reuters) Reuters - Short-sellers, Wall Street's dwindling of ultra-cynics, are seeing green again. | Business |
| **Emotions** | i feel romantic too. | Love |
| **Dbpedia-14** | Abbott of Farnham E D Abbott Limited was a British coachbuilding business based in Farnham Surrey trading under that name from 1929. A major part of their output was under sub-contract to motor vehicle manufacturers. Their business closed in 1972. | Company |

Table 33: Samples from Standard Text Classification Benchmarks

| Type | Context Sample | Target |
|---|---|---|
| ***Pronoun Resolution*** | | |
| | Story highlights Authorities also threaten to arrest Musharraf if | he |
| | Myriam Steinberg was 40 and fresh off a breakup when | she |
| | Following the airport shooting, Steube doubled down on | his |
| | I can't express how extremely pleased we are with Jenny and | her |
| | From 1 July 2018, the Tax Office is advising Australians that if | they |
| | No other appliance company has a wider scope of solutions, nor the experience to back | them |
| ***Delimiters(Brackets)*** | | |
| | Q: Is the sum of separating vectors always separating? If $\mathcal | { |
| | Let w be u | ( |
| | —abstract: 'Franson's Bell experiment with energy-time entanglement | [ |
| ***Punctuation*** | | |
| | During my pregnancy, I tried to gather as much information on how painful labor might actually be | . |
| | We are excited that you will be attending the Madison Mission Trip | ! |
| | Ever noticed how close ribbit and rabbit sounded | ? |
| ***Subject-Verb Agreement*** | | |
| | In physics, the fundamental interactions, also known as fundamental forces, are the interactions that | do |
| | Asexual reproduction is a type of reproduction that | does |
| | An allocution, or allocutus, is a formal statement made to the court by the defendant who | has |
| | The Abencerrages or Abencerrajes (from the Arabic for Šaddler's Son) were a family or faction that is said to | have |

Table 34: Samples for Fine-Grained Concept Completion

