# OpenReview forum: "Neurons Speak in Ranges: Breaking Free from Discrete Neuronal Attribution"
_TMLR — Accepted by TMLR_

### Review · Reviewer_5PyT · 2026-02-02

**Summary Of Contributions:**

In this work, the authors present observations regarding neuron polysemanticity—the tendency of individual neurons to activate in response to multiple, seemingly unrelated concepts. Through statistical analysis of a concept-level annotated dataset, they report that the activation distributions of these neurons exhibit Gaussian-like properties. Leveraging this observation, they define concepts as ranges capturing the bulk of the activation distribution and introduce NeuronLens, a neuron interpretation method. The authors position NeuronLens as an alternative to existing masking-based approaches, claiming improved interpretability.

Strengths:

- The paper addresses a highly relevant and valuable topic: the analysis of patterns in the functioning of pre-trained networks. The statistical analysis section presents a simple yet insightful observation that warrants attention and further exploration.
- The breadth of experimental effort is commendable, with evaluations conducted across three models and three datasets, supplemented by extensive ablation studies. The inclusion of detailed versions of existing tables in the appendix further enhances the reproducibility and transparency of the work.
- Additionally, the visualization of neuronal activation patterns is effective

Weaknesses:

- Overall, the paper suffers from issues related to clarity and rigor. While some of the concerns are minor and do not fundamentally undermine the integrity of the study, others significantly hinder the comprehension of the presented work (see the "Requested Changes" section for specific examples).
- The experimental validation provided to justify the performance of the NeuronLens method is somewhat unsatisfactory. In my view, the default approach for assessing performance in neuron identification experiments should involve mean replacement, as the objective should be to neglect the neuron’s effect rather than to "attack" it. Although the authors address this point, their response is limited to a specific setup where accuracy reaches 1 and disturbed accuracy drops to -1 for most classes. More broadly, many experimental setups yield curious results, with numerous instances of below-zero accuracies, which raises questions about the robustness and generalizability of the findings.
- Upon examining Figure 4, it appears that while NeuronLens identifies neurons associated with a substantial drop in accuracy for the relevant concept compared to neuron masking, it also causes greater deterioration in auxiliary concepts.
- Additionally, NeuronLens operates by probing each neuron individually. However, there is strong reason to believe that polysemanticity should be interpreted as arbitrary directions in the latent space [1]. As such, I would find it more compelling to frame the method as neuron-independent—for instance, by searching for concept representations as multivariate Gaussian distributions within the latent space.

**Additional Comments:**

[1] Elhage, Nelson, et al. "Toy models of superposition." arXiv preprint arXiv:2209.10652 (2022).

[2] Cunningham, Hoagy, et al. "Sparse autoencoders find highly interpretable features in language models." arXiv preprint arXiv:2309.08600 (2023).

[3] Nostalgebraist. "Interpreting GPT: The logit lens." Blog Post (2020).

**Audience:**

Yes

**Audience Explanation:**

Definitely yes, the study of neuron polysemanticity is undeniably a topic of significant current interest in the field. This paper offers compelling and insightful observations that contribute to a deeper understanding of the phenomenon.

**Claims And Evidence:**

No

**Claims Explanation:**

The statistical analysis presented in the manuscript is, in my assessment, sufficiently supported by the evidence. However, I remain unconvinced by the claims regarding the superiority of NeuronLens. For further details, please refer to the "Weaknesses" section of this review.

**Requested Changes:**

Minor:

- The metric for accuracy is presented as a percentage in Figure 4 but scaled between -1 and 1 in the tables. It would be advisable to standardize the representation.

- There is notable redundancy in several statements. For example: "However, neurons frequently exhibit polysemanticity; the ability to encode multiple, seemingly unrelated concepts (Lecomte et al., 2024; Marshall & Kirchner, 2024)" vs. "Work in polysemanticity (Lecomte et al., 2024; Marshall & Kirchner, 2024) shows that individual neurons learn multiple seemingly unrelated concepts." Another example:  "Related Works" section overlaps with content in Paragraph 2 of the Introduction.

- Descriptions of experimental results are occasionally verbose without adding substantive information. For instance: "Full neuron masking leads to notable increases in perplexity, exceeding 3 points in the best case, whereas range masking results in a maximum increase of only 1.1."

- Typos: lines under Table 4 are misaligned; a reference to the appendix is missing in Table 2; Sentence "Figures figs. 8 to 19 in Figure 4" in section F.2 unclear

- Consider adding arrows (↑/↓) in tables to indicate whether higher or lower values are better. Some best results are not bolded for emphasis, while others are (Example Table 3 or Table 5).

- The Related Works section could be strengthened. Even if not directly related to NeuronLens, it would be valuable to discuss Sparse Autoencoders [2] and LogitLens [3], given their methodological similarities to the proposed approach.

Major:

- Provide a clearer definition of what distinguishes a salient neuron from a non-salient one. The current approach appears to rely on thresholding maximum activation values, but the choice of thresholds (50%, 30%, 20%, and 5%) lacks justification and seems arbitrary.

- Explicitly define what constitutes a concept in the experiments. Is it associated with a token (as suggested by probability distribution plots), a word (as implied by Table 2), or sentences (as indicated by Table 7)?

- Include mean replacement as a baseline in the experimental evaluations to facilitate more robust comparisons.

---

> ### Author Response · Authors · 2026-02-20
> **Response to Reviewer[Part 1].**
>
> **The experimental validation provided to justify the performance of the NeuronLens method is somewhat unsatisfactory. In my view, the default approach for assessing performance in neuron identification experiments should involve mean replacement, as the objective should be to neglect the neuron’s effect rather than to "attack" it. Although the authors address this point, their response is limited to a specific setup where accuracy reaches 1 and disturbed accuracy drops to -1 for most classes. More broadly, many experimental setups yield curious results, with numerous instances of below-zero accuracies, which raises questions about the robustness and generalizability of the findings.**
>
> We apologize for the confusion. The base accuracies are 1 for experiments conducted in some settings, as filtering was performed to include only samples where the base model was able to predict correctly (Noted in Appendix H).  The negative numbers are not scaled accuracies but represent a change in performance, which in this case denotes the drop in performance. We have added deltas to all tables to showcase that these are drops in accuracy and confidence.
>
> **Upon examining Figure 4, it appears that while NeuronLens identifies neurons associated with a substantial drop in accuracy for the relevant concept compared to neuron masking, it also causes greater deterioration in auxiliary concepts.**
>
> We apologize for the confusion. In Figure 4 (and across all experimental settings), we select the same neurons to be ablated by either full neuron intervention or range-based intervention. In Figure 4, the solid lines show the drops in the targeted concepts. Here, we observe virtually the same deterioration between full neuron ablation and range-based ablation. The dotted lines in the figure show the collateral effect on other concepts. We see less damage in the case of ablating the neuron range in comparison to full neuron ablation.
>
> **Additionally, NeuronLens operates by probing each neuron individually. However, there is strong reason to believe that polysemanticity should be interpreted as arbitrary directions in the latent space [1]. As such, I would find it more compelling to frame the method as neuron-independent—for instance, by searching for concept representations as multivariate Gaussian distributions within the latent space.**
>
> We agree that concepts are often represented as combinations of neurons (directions) rather than single neurons. We added new activation-steering results, including a range-gated steering variant, to show our approach applies to direction-based interventions as well. Please refer to the reviewer **mxe6** response for the activation steering experimental details.
>
> **There is notable redundancy in several statements. For example: "However, neurons frequently exhibit polysemanticity; the ability to encode multiple, seemingly unrelated concepts (Lecomte et al., 2024; Marshall & Kirchner, 2024)" vs. "Work in polysemanticity (Lecomte et al., 2024; Marshall & Kirchner, 2024) shows that individual neurons learn multiple seemingly unrelated concepts." Another example: "Related Works" section overlaps with content in Paragraph 2 of the Introduction.**
>
> We thank the reviewer for highlighting redundant statements. We have removed the text from Section 3 (Polysemanticity) as the information was covered in the introduction.
>
> **Descriptions of experimental results are occasionally verbose without adding substantive information.**
>
> We thank the reviewer for flagging this. We updated the referenced text, and if we identify additional instances, we will update them in the camera-ready version.
>
> **Typos: lines under Table 4 are misaligned; a reference to the appendix is missing in Table 2; Sentence "Figures figs. 8 to 19 in Figure 4" in section F.2 unclear**
>
> We thank the reviewer for pointing this out. We have updated the text, which now reads as “Figures 8 to 19“.
>
> **Consider adding arrows (↑/↓) in tables to indicate whether higher or lower values are better. Some best results are not bolded for emphasis, while others are (Example Table 3 or Table 5).**
>
> Thank you for the suggestion. We have added ∆ (delta) columns throughout the tables to make improvements and degradations explicit. We chose not to include ↑/↓ indicators because, given the number of metrics and columns, they substantially increase visual clutter and reduce readability. Instead, we state in the Metrics section whether higher or lower is better for each metric, and we ensure the tables follow that convention consistently.
>
>
> **The Related Works section could be strengthened. Even if not directly related to NeuronLens, it would be valuable to discuss Sparse Autoencoders [2] and LogitLens [3], given their methodological similarities to the proposed approach.**
>
> We have revised the related section to include the suggested references and other related works.

---

> ### Author Response · Authors · 2026-02-20
> **Response to Reviewer[Part 2].**
>
> **Provide a clearer definition of what distinguishes a salient neuron from a non-salient one. The current approach appears to rely on thresholding maximum activation values, but the choice of thresholds (50%, 30%, 20%, and 5%) lacks justification and seems arbitrary.**
>
> Our definition of salient neurons relies on existing neuron-concept identification methods to select salient neurons. In the paper, we first empirically evaluate three neuron ranking approaches and select maximally activating ranking as it produces the largest targeted drop while preserving complementary concepts. For detailed results, please refer to Appendix C.
>
> For the percentage of neurons to ablate, since in the mechanistic interpretability and concept ablation literature, there is no universally established, fixed percentage for neuron masking, as the optimal threshold depends entirely on the model architecture, the dataset, and the specific concept's distribution [1] [2]. Standard practice dictates determining this threshold empirically. We first test different percentages on our largest dataset, DB-14. We find that in the case of pre-trained llama model, it removes the majority of the target concept >98%, with 30% neurons.
>
> For the GPT model, we provide ablation from 10-90% of neurons selected in the main text Figure 4. We presented the results for 50% on this model in the main text, and for consistency, we also ablated the same percentage of neurons in the Bert-based models.
>
> We used the top 5% for qualitative analysis to focus on the extreme cases while keeping the number of neurons small enough for manual inspection. For example, in the experiment for Figure 7, we only find 7 neurons that were in the salient set for all 14 classes, when 5% highest activating neurons were selected. We show the box plots of these neurons to show that even these highly polysemantic neurons have separation between concepts.
>
> Importantly, across all the percentages ablated, in all settings, and even without selecting any neurons and ablating the full feature vector (example reviewer **mxe6**  steering experiments), our approach substantially outperforms full neuron ablations.
>
> [1] Transformer Feed-Forward Layers Are Key-Value Memories
>
> [2] Understanding the Role of Individual Units in a Deep Neural
>
> **Explicitly define what constitutes a concept in the experiments. Is it associated with a token (as suggested by probability distribution plots), a word (as implied by Table 2), or sentences (as indicated by Table 7)?**
>
> We have added details about concepts in tables 31 and 32 to show concept samples and labels. The concepts used in the main text are sequence-level: they are categorical labels assigned to entire samples by human annotators during dataset curation. For example, in the AG News dataset, each article is labeled as one of four classes: World, Sports, Sci/Tech, or Business. We have conducted further experiments for finer concepts. Please refer to the central response above for the results.
>
> **Include mean replacement as a baseline in the experimental evaluations to facilitate more robust comparisons.**
>
> We thank the reviewer for their valuable suggestion. We have added results for mean ablation on all three datasets for the Llama setting, presented in Tables 12 - 14. Across metrics (Acc, Conf, CAcc, and CConf), activation range masking consistently outperforms neuron masking. The degradation in accuracy and confidence of auxiliary concepts is significantly lower under range-based interventions. Additionally, we observe substantial differences in how the two masking methods affect general language modelling capabilities. Neuron masking leads to a large increase in perplexity (PPL), ranging from +7.413 to +23.25 in comparison to range masking, where the increase in perplexity is limited to +0.687 at max. Here we provide averaged results from these experiments.
>
> *AG News*
> | Method | $\Delta$Acc | $\Delta$Conf | $\Delta$CAcc | $\Delta$CConf | $\Delta$PPL |
> |---|---:|---:|---:|---:|---:|
> | Neuron Masking | -0.983 | -0.509 | -0.376 | -0.481 | 22.43 |
> | Activation Range Masking | -0.950 | -0.506 | **-0.329** | **-0.395** | **0.431** |
>
> *Emotions*
> | Method | $\Delta$Acc | $\Delta$Conf | $\Delta$CAcc | $\Delta$CConf | $\Delta$PPL |
> |---|---:|---:|---:|---:|---:|
> | Neuron Masking | -0.863 | -0.516 | -0.665 | -0.449 | 7.09 |
> | Activation Range Masking | -0.813 | -0.515 | **-0.616** | **-0.423** | **0.501** |
>
> *DB-14*
>
> | Method | $\Delta$Acc | $\Delta$Conf | $\Delta$CAcc | $\Delta$CConf | $\Delta$PPL |
> |---|---:|---:|---:|---:|---:|
> | Neuron Masking | -0.985 | -0.540 | -0.632 | -0.546 | 7.899 |
> | Activation Range Masking | -0.984 | -0.539 | **-0.408** | **-0.498** | **0.508** |

---

> > ### Comment · Reviewer_5PyT · 2026-02-23
> >
> > I thank the authors for their thorough and detailed rebuttal. The amount of additional work provided is appreciated, and I found the revised results substantially clearer. I consider that nearly all points of tension have been adequately addressed.
> >
> > However, one aspect remains not entirely clear to me, namely the effect of the deterioration on the target concept $c$ (presented in the rebuttal through $\Delta acc$ and $\Delta conf$). In my view, the expected behaviour of an effective manipulation framework should result in the highest drop in accuracy and confidence, since the goal is to negate a supposedly meaningful concept. Yet the largest drop appears to occur in the baseline.
> >
> > In light of this, I would like to request the following clarifications:
> >
> > - How do the authors interpret the $\Delta acc$ and $\Delta conf$ results? Theoretically, what should the best expected behaviour be? This is particularly puzzling given that the results in these columns appear to not be bolded.
> > - The main remark regarding these results is that "we observe virtually the same deterioration." While this seems to hold in some cases, I find it difficult to draw the same conclusion when the resulting accuracy is close to zero: a drop between 0.95 and 0.90 is not comparable to a drop between 0.10 and 0.05, as the latter is far less noticeable. A typical example is the first plot of Figure 4 in the high-percentage regime. I would welcome the authors' perspective on this point.
> > - A straightforward way to reduce collateral degradation is simply to perform less aggressive neuron masking, which may also result in smaller $\Delta acc$, $\Delta conf$, $\Delta C acc$ and $\Delta C conf$. To what extent does the proposed method differ from this simpler strategy in terms of results?

---

> ### Author Response · Authors · 2026-02-24
> **Response to Reviewer.**
>
> We thank the reviewer for their time and energy and appreciate the constructive and detailed feedback they have provided throughout the reviewing cycle.
>
> **How do the authors interpret the $\Delta acc$ and $\Delta conf$ results? Theoretically, what should the best expected behaviour be? This is particularly puzzling given that the results in these columns appear to not be bolded.**
>
> **The main remark regarding these results is that "we observe virtually the same deterioration."...A typical example is the first plot of Figure 4 in the high-percentage regime. I would welcome the authors' perspective on this point.**
>
>
> The goal is to not only negate a target concept but also to have minimal collateral impact on complementary concepts. In all settings, we boldface only the metrics for complementary concepts. For the targeted concepts, we aim to keep the induced drops comparable across methods. We do expect the target concept accuracy to drop slightly less (by a small ε) in the activation-range masking setting: masking ±2.5 standard deviations removes roughly 98.76% of the mass under a Gaussian assumption, so a small fraction of activations still passes through. In practice, however, the drops on the targeted concepts remain comparable across methods and settings, while our approach better preserves performance on the complementary concepts. Regarding Figure 4, the largest gap in targeted deterioration between the two approaches is about 3.5 percentage points, occurring at the highest ablation rate on DBpedia, whereas the complementary concepts exhibit a substantially larger separation.
>
> To better showcase the gains of our method, one can report a relative-drop metric, for example, the gap between the main-concept drop and the complementary-concept drop (ΔAcc − ΔCAcc), and compare this quantity across methods to highlight the benefit of our approach.
>
>
> | Dataset    | $\Delta$Acc-$\Delta$CAcc Neuron Masking | $\Delta$Acc-$\Delta$CAcc Activation Range Masking |
> |-|-|-|
> | AG News|27| **45**|
> | Emotions|10| **20**|
> | DBpedia-14| 9| **32**|
>
> We considered a metric of this form, but ultimately chose not to include it because it abstracts away one of the core goals of this work: providing a causal demonstration that the targeted concepts reside in thin slices of activation space. For this reason, we report the raw drops in targeted concept (and complementary concepts) accuracy. Showing that masking most of the identified activation range produces a comparable effect on the targeted concept, providing concrete evidence that the concept is indeed encoded within the observed activation range. In contrast, relative summary metrics can obscure this causal link.
>
> Additionally, a setting with a higher number of neurons ablated would outperform full neuron masking for both Acc and CAcc (as can be observed from the results below), but this would not communicate where the targeted concept is localized or why the intervention works. For this reason, we showcase that if we ablate the same neurons of the model, with the only difference of either masking an activation range or the full neuron.
>
>
> **A straightforward way to reduce collateral degradation is simply to perform less aggressive neuron masking, which may also result in smaller...**
>
>
> If we aim to reduce collateral damage under full-neuron masking, we must substantially reduce the number of neurons we ablate, which in turn yields less deterioration on the targeted concept. To illustrate this tradeoff, we ablate full-neuron masking with reduced neuron counts to approximately match the drop in complementary concepts with activation range masking.
>
>
> *AG News:*
>
> In the case of AG_News, we can ablate up to 50 percent of neurons using activation range masking, while producing similar drops in complementary concepts to full neuron masking at 10%.
>
> |Method| $\Delta$Acc | $\Delta$CAcc |
> |-|-|-|
> |Neuron Masking(10%)| -26%| -11%|
> |Activation Range Masking(50%) |-93%| -11%|
>
>
> *DBpedia-14:*
>
> In the extreme case of DBpedia, we can ablate up to 90% of neurons in comparison to 10 percent of neurons ablated using full neuron making, with similar drops in complementry concepts.
>
>
> | Method|$\Delta$Acc|$\Delta$CAcc |
> |-|-|-|
> |Neuron Masking(10%)|-10%|-6%|
> |Activation Range Masking(50%)|-96%| -5%|
>
>
> *Emotions:*
>
> |Method|$\Delta$Acc|$\Delta$CAcc|
> |-|-|-|
> |Neuron Masking(30 %)|-83%|-19%|
> |Activation Range Masking(50 %)| -92%| -19%|
>
> *All results rounded off to full percentage point for clarity.
>
> Across all experiments, we observe the same qualitative pattern: matching complementry preservation requires full-neuron masking to operate on far fewer neurons, whereas activation-range masking can act on substantially larger neuron sets while producing consistently larger targeted-concept drops. This supports the central claim of the paper that activation-range masking yields more faithful and precise attributions.

---

> > ### Comment · Reviewer_5PyT · 2026-02-24
> >
> > I thank the authors for their prompt and extensive rebuttal, which clarified the remaining points and helped me better understand the authors' perspective. Concerning my case, I have no further concerns or questions. I believe all issues raised in my review have been adequately addressed.

---

### Review · Reviewer_mxe6 · 2026-02-07

**Summary Of Contributions:**

The paper argues that despite widespread polysemanticity, individual neurons in LLMs exhibit concept-conditioned activation ranges that are approximately Gaussian and often separable, enabling more precise interpretation and control than discrete neuron-to-concept mappings. It introduces NeuronLens, which estimates per-concept activation ranges per neuron and applies range-gated interventions only within those ranges. Across multiple models and datasets, range-based edits suppress target concepts with substantially less collateral damage to auxiliary concepts than whole-neuron masking, while better preserving general capabilities.

Strength:
1. The paper is well-written and easy to follow.
2. The proposed method is well-motivated and is analyzed comprehensively.
3. The numerical experiments show promising performance of NeuronLens.

Weaknesses:
1. The paper omits discussion of sparse autoencoders (SAEs), a prominent approach to addressing polysemanticity. It would help to situate NeuronLens relative to SAEs. In SAE-based views, disentangled concept features are represented as sparse linear combinations of neuron activations. By contrast, NeuronLens operates at the level of individual coordinates, attributing concepts to neuron-specific activation ranges. One way to reconcile these views is to note that range-based attribution can be seen as examining the per-neuron projections of a concept-direction: different concept directions induce different coordinate-wise activation distributions, which manifest as distinct ranges.
2. Evaluations center on class-level semantics. The concept encoded by neurons are usually finer-grained. It is unclear whether the separability assumptions and the NeuronLens also work for these finer-grained concepts.
3. Comparisons focus on neuron masking; stronger baselines (direction/subspace editing, causal tracing variants) would better situate gains. Also the authors can consider evaluating the method on SAE related benchmarks.

**Audience:**

Yes

**Audience Explanation:**

The paper offers a fresh coordinate-level perspective on polysemanticity, together with a practical framework, NeuronLens, for concept intervention. These findings are interesting to researchers in interpretability, controllability, and safety of LLMs.

**Claims And Evidence:**

Yes

**Claims Explanation:**

1. The claim that concept-conditioned activations often form approximately Gaussian, separable ranges within a neuron is supported by consistent KDE visualizations, skewness/kurtosis near-normality, and a practical-normality KS criterion across several models and datasets.
2.  The effectiveness of NeuronLens is proven by numerical experiments. The results show effective targeted suppression with notably less collateral degradation than whole-neuron masking.

**Requested Changes:**

1. Clarifying how range-gating complements or differs from SAE, e.g., when coordinate-level gating is preferable to editing along learned feature directions.
2. Adding some experiments regarding intervention of finer-grained concepts.
3. Adding direction/subspace editing comparators and a causal-tracing style targeted intervention where feasible.
4. It will make the method of higher practical value if the authors could provide some (heuristic) algorithms for selecting the optimal $\tau$ (besides extensive hyperparameter tuning).

---

> ### Author Response · Authors · 2026-02-20
> **Response to Reviewer.**
>
> **Clarifying how range-gating complements or differs from SAE...**
>
> We have added a discussion in the related work section. Moreover, we have conducted preliminary experiments on concept removal in SAE features, comparing full feature masking with activation range masking. For experimental details and additional discussion, please refer to our central response.
>
> **Adding some...finer-grained concepts.**
>
> We thank the reviewer for their valuable suggestion. We have provided experiments for additional finer concepts in the central response above. Please refer to those and Appendix K for further details in the paper document.
>
> **Adding direction/subspace editing comparators and a causal-tracing style targeted intervention where feasible.**
>
> Thank you for suggesting additional baselines. We have provided results for direction ablation and activation steering methods in Appendix M. Here, we provide averaged results from the experimentation:
>
> *Activation Steering*
>
> We first compare activation range masking with zero replacement with mid-depth activation steering[1] (Steering applied at the middle layer of the model).
>
> |Method|$\Delta$Acc|$\Delta$Conf|$\Delta$CAcc|$\Delta$CConf|$\Delta$PPL|
> |-|-:|-:|-:|-:|-:|
> |Activation Steering|-0.75|-0.33|**-0.140**|**-0.027**|1.11|
> | Activation Range Masking |-0.94|-0.44|-0.301|-0.261|**0.655**|
>
> From the results, we observe that our approach is able to produce higher removal of the target concept, with substantially lower perplexity increase. Interestingly, activation steering completely fails to remove one of the concepts, specifically the concept of world news. Which indicates that a clear steering direction does not exist for this concepts. Additionally, standard steering relies on both a target and an auxiliary dataset to determine direction; our method simplifies this by requiring only the target concept dataset.
>
> *Directional Ablation*
>
> We then compare the same setting with projection removal [2] from the mid-depth layer of the model. We find that directional ablation completely fails in terms of targeted removal; it removes the main concept along with substantial drops in complementary concepts, and produces substantially higher perplexity.
>
> |Method|$\Delta$Acc|$\Delta$Conf|$\Delta$CAcc|$\Delta$CConf|$\Delta$PPL|
> |-|-:|-:|-:|-:|-:|
> |Direction Ablation|-1.0|-0.514|-0.972|-0.512|994.47|
> |Activation Range Masking|-0.94|-0.44|**-0.301**|**-0.261**|**0.655**|
>
> *Range-based Steering*
>
> To gauge the combined effect and usefulness of activation range along with activation steering, we apply activation steering in our default setting (where we find direction based on the last contextualized token instead of the average representation). We apply this version of activation steering to the last layer of the model.
> |Method|$\Delta$Acc|$\Delta$Conf|$\Delta$CAcc|$\Delta$CConf|$\Delta$PPL|
> |-|-:|-:|-:|-:|-:|
> |Activation Steering|-0.984|-0.495|**-0.243**|-0.021|4.099|
> |Activation Range (Activation Steering)|-0.975|-0.492|-0.259|**-0.020**|**1.323**|
>
> We find that activation steering on the last contextualized token is able to remove the targeted concept precisely in this setting while substantially conserving the complementary concepts, but at a cost of higher perplexity. We then compare this setting with our range-based activation steering, i.e., only activation steer when the activation of the coordinate falls in the activation range for the concept. We find that in this setting, the performance is similar to the baseline activation steering, but the perplexity is substantially reduced, highlighting the complementary importance of our approach with other methods.
>
> We do not include causal tracing in our evaluation due to computational overhead and time constraints.
>
> **It will make the method of higher practical ...**
>
> For the experiments in the main paper, we utilize $\tau$ set to 2.5 to enable a direct comparison with full neuron masking, ensuring that auxiliary gains are reported on a comparable scale. We have provided an alternative method, adaptive dampening in Appendix G.3 (Alternative Activation Interventions), which removes the need to search for an optimal value. In this method, we dampen the activation of the neuron based on how centered it is to the distribution’s center. This method modulates suppression in proportion to each activation’s deviation from its class-conditional mean, enabling suppression guided by sampled observations. Adaptive dampening achieves the strongest balance across all metrics: perplexity remains low (0.41–0.61), MMLU is maintained or improved (up to +0.03), and collateral damage to auxiliary concepts is minimized (CAcc drops consistently below -0.3, often under -0.15), outperforming dampening, mean replacement, and zeroing out approaches.
>
> [1] Persona Vectors: Monitoring and Controlling Character Traits in Language Models.
>
> [2] Refusal in Language Models Is Mediated by a Single Direction.

---

### Review · Reviewer_EVJs · 2026-02-09

**Summary Of Contributions:**

This paper shows that many “salient” neurons in LLMs are polysemantic, but the activation magnitudes associated with different concepts often form distinct, roughly Gaussian-like distributions with little overlap. The authors then propose a interpretability method NeuronLens, which attributes concepts to activation _ranges_ within a neuron and uses range-based interventions to manipulate a target concept while causing less collateral damage than masking whole neurons  . Experiments and causal “concept erasure” evaluations show that range-based interventions can manipulate target concepts while causing substantially less collateral damage to other concepts and overall model performance than full neuron masking.

**Audience:**

Yes

**Audience Explanation:**

This paper tackles the problem of interpretability in Large Language Models. I think this should be of interest to the general TMLR community.

**Broader Impact Concerns:**

I do not have broader impact concern on this paper.

**Claims And Evidence:**

Yes

**Claims Explanation:**

1. It would be good to provide more evidence on the core assumption (“Gaussian-like, low-overlap concept-conditioned distributions”). For example, a quantitative overlap measure across concepts per neuron (e.g., overlap coefficient / AUROC between concept-conditioned distributions). It would also be good to show the fraction of neurons where overlap is small enough to make range-gating meaningful.
2. For neuron activations, it would be better to provide a breakdown by layer/model/dataset. Also document clear failure cases (high-overlap or multimodal).
3. The paper is well written with clear motivation and the authors have conducted detailed and comprehensive experiment analysis. In addition to GPT-2, the authors conduct analysis on recent LLMs, i.e., Llama 3 model series.

**Requested Changes:**

1. In section 2.1, can the authors provide more details on the definition of "Concept"? For example, why is there only four concept type for a sentence? Can the authors provide concrete examples of concepts.
2. I think the related work could be made to be more complete. There are many works on mechanistic interpretability that tries to tackle similar problems, it would be good to add more discussion on related works and highlight the novelty/contributions of this work as compared to others.
3. Minor but there are some typos: in the caption of table 2 "full reference texts are provided in Appendix ??".
4. It would be good to clarify the full neuronlens pipeline end-to-end with algorithmic detail, for instance, pseudocode would be helpful for demonstrating the method better.
5. How is the proposed method Neurolens related to sparse auto-encoders (SAEs)? SAEs also seem to be a popular approach for LLM interpretability. It would be good to add some discussions on this in the paper.

---

> ### Author Response · Authors · 2026-02-20
> **Response to Reviewer.**
>
> **It would be good to provide more evidence on the core assumption (“Gaussian-like, low-overlap concept-conditioned distributions”). For example, a quantitative overlap measure across concepts per neuron (e.g., overlap coefficient / AUROC between concept-conditioned distributions). It would also be good to show the fraction of neurons where overlap is small enough to make range-gating meaningful. For neuron activations, it would be better to provide a breakdown by layer/model/dataset. Also document clear failure cases (high-overlap or multimodal).**
>
> Thank you for the suggestion. We include statistics regarding neuron activation overlap for various concepts using AUROC, considering all possible pairwise combinations of concepts within datasets. Due to computational overhead, we provide these statistics for the last layer of the models only. We have included these statistics in the Appendix section D.2 (Quantitative).
> | Model | Dataset | Q1 (< 25%) | Q2 (25% - 50%) | Q3 (50% - 75%) | Q4 (>= 75%) |
> | :--- | :--- | :--- | :--- | :--- | :--- |
> | Bert | DB-14 | 85.90% | 6.15% | 4.19% | 3.76% |
> | Bert | Emotions | 56.11% | 17.87% | 14.18% | 11.83% |
> | Bert | AG_News | 52.26% | 20.01% | 14.80% | 12.93% |
> | GPT-2 | DB-14 | 66.70% | 13.53% | 10.39% | 9.38% |
> | GPT-2 | Emotions | 21.95% | 25.72% | 26.46% | 25.87% |
> | GPT-2 | AG_News | 22.94% | 26.97% | 25.74% | 24.35% |
> | Llama-3 3.2B | DB-14 | 26.75% | 25.41% | 24.17% | 23.67% |
> | Llama-3 3.2B | Emotions | 4.41% | 19.72% | 34.13% | 41.74% |
> | Llama-3 3.2B | AG_News | 17.93% | 26.46% | 27.53% | 28.09% |
> | Llama-3 3.2B | All Data | 43.38% | 20.15% | 18.46% | 18.01% |
>
> In the finetuned settings, we observe minimal overlap between concepts. In the pretrained setting, we observe more overlap. We observe the highest overlap in the pretrained emotions classification setting, which is in line with our findings in the main paper Table 4, where we see the limited gains in this setting from using our approach.
>
> Though we note that these overlaps are between the concepts tested, there are possibly many other concepts that are preserved using our approach, as evidenced by the perplexity comparison, where our method substantially outperforms full neuron masking across all settings.
>
> To further illustrate this point, we concatenate all concepts from all datasets in the Llama setting. Since all concepts tested are functions of the same underlying model weights, when we check the overlap ratios again in this setting, we find that, as we consider more concepts, the overlap ratios substantially decrease.
>
> **In section 2.1, can the authors provide more details on the definition of "Concept"? For example, why is there only four concept type for a sentence? Can the authors provide concrete examples of concepts.**
>
> The concepts used in the main text are sequence-level: they are categorical labels assigned to entire samples by human annotators during dataset curation. For example, in the AG News dataset, each article is labeled as one of four classes: World, Sports, Sci/Tech, or Business. We have conducted further experiments for finer concepts. Please refer to the central response above for the results. Generally, a concept refers to a human-interpretable feature or category. We have added details about concepts in Tables 31 and 32 of the paper document to show concept samples and labels.
>
> **I think the related work could be made to be more complete. There are many works on mechanistic interpretability that tries to tackle similar problems, it would be good to add more discussion on related works and highlight the novelty/contributions of this work as compared to others.**
>
> We thank the reviewer for their suggestion. We have expanded the related work section in the paper.
>
> **Minor but there are some typos: in the caption of table 2 "full reference texts are provided in Appendix ??".**
>
> We thank the reviewer for pointing out the missing reference. The missing reference is to Appendix E (Qualitative Text Analysis) Table 7. We have updated the table caption.
>
> **It would be good to clarify the full neuronlens pipeline end-to-end with algorithmic detail, for instance, pseudocode would be helpful for demonstrating the method better.**
>
> We have added the algorithm for the overall pipeline in Appendix L (Range Masking Algorithm).
>
> **How is the proposed method Neurolens related to sparse auto-encoders (SAEs)? SAEs also seem to be a popular approach for LLM interpretability. It would be good to add some discussions on this in the paper.**
>
> We have added a discussion in the related work section. Moreover, we have conducted preliminary experiments on concept removal in SAE features, comparing full feature masking with activation range masking. For experimental details and additional discussion, please refer to our central response.

---

### Author Response · Authors · 2026-02-20
**Central Response Concepts.**

**Reviewer EVJs: In section 2.1, can the authors provide more details on the definition of "Concept"? For example, why is there only four concept type for a sentence? Can the authors provide concrete examples of concepts.**

**Reviewer Mxe6: Adding some experiments regarding the intervention of finer-grained concepts.**

**Reviewer 5PyT: Explicitly define what constitutes a concept in the experiments. Is it associated with a token (as suggested by probability distribution plots), a word (as implied by Table 2), or sentences (as indicated by Table 7)?**


We consider a broad definition of a concept that may be coarse-grained (e.g., a dataset class label) or fine-grained (e.g., a lexical or syntactic pattern defined over tokens). This definition lets us study concept-conditioned activations across multiple levels of abstraction. In the main experiments, concepts are sequence-level and correspond to categorical labels assigned to entire samples during dataset curation. For example, in AG News, each sample is labeled as one of four classes: World, Sports, Sci/Tech, or Business, and we treat each class as a concept. Beyond these class-level concepts, we provide additional evaluations on finer-grained concepts, such as pronouns[1], delimiters (brackets) [3], terminal punctuation [1,2], and subject-verb agreement [1,4] (details are provided below). These finer concepts include predefined token sets or token patterns. Throughout our experimentation, our goal is to study how neuron activations behave when conditioning on a concept of interest and whether targeted interventions can selectively affect that concept without significant collateral.

We provide samples for concepts in Tables 31 and 32 of the paper document.

Concept Details: For pronouns, we consider masculine, feminine, and neutral forms: he/his, she/her, and they/them. For terminal punctuation, we analyze periods (.), question marks (?), and exclamation marks (!). For delimiters, we consider the opening braces \{, (, and [, and for subject–verb agreement, we analyze both singular and plural verb forms, specifically do/does, and has/have.

We report the mean performance across tasks below. Overall, these results echo the main document’s experiments: complementary concepts are better preserved, while degradation of the targeted concept is comparable to that achieved with neuron masking. This suggests that activation range is a general-purpose approach that transfers across a broad set of concepts. Full per-task breakdowns for finer-grained concepts are available in the paper; we refer the reader to Appendix K for details.

*Gender Pronouns:*

| Method | $\Delta$Acc | $\Delta$Conf | $\Delta$CAcc | $\Delta$CConf | $\Delta$PPL |
|---|---:|---:|---:|---:|---:|
| Neuron Masking | -0.73 | -0.45 | -0.24 | -0.13 | 2.03 |
| Activation Range Masking | -0.73 | -0.43 | **-0.06** | **0.06** | **0.56** |

*Termination Tokens:*

| Method | $\Delta$Acc | $\Delta$Conf | $\Delta$CAcc | $\Delta$CConf | $\Delta$PPL |
|---|---:|---:|---:|---:|---:|
| Neuron Masking | -0.72 | -0.47 | -0.46 | -0.28 | 1.49 |
| Activation Range Masking | -0.66 | -0.41 | **-0.14** | **-0.05** | **0.34** |

*Delimiters:*

| Method | $\Delta$Acc | $\Delta$Conf | $\Delta$CAcc | $\Delta$CConf | $\Delta$PPL |
|---|---:|---:|---:|---:|---:|
| Neuron Masking | -0.70 | -0.48 | -0.24 | -0.25 | 1.69 |
| Activation Range Masking | -0.68 | -0.47 | **-0.09** | **-0.08** | **0.41** |

*Subject Verb Agreement:*
| Method | $\Delta$Acc | $\Delta$Conf | $\Delta$CAcc | $\Delta$CConf | $\Delta$PPL |
|---|---:|---:|---:|---:|---:|
| Neuron Masking | -0.66 | -0.34 | -0.27 | -0.10 | 1.33 |
| Activation Range Masking | -0.70 | -0.33 | **-0.10** | **0.02** | **0.34** |

[1] Finding Neurons in a Haystack: Case Studies with Sparse Probing. TMLR 2023

[2] Punctuation and predicates in language models. Arxiv 2025

[3] Language models make balanced parentheses errors when faulty mechanisms overshadow sound ones. COLM 2025

[4] Sparse feature circuits: Discovering and editing interpretable causal graphs in language models. ICLR 2025

---

### Author Response · Authors · 2026-02-20
**Central Response SAEs.**

**Reviewer EVJs: How is the proposed method Neurolens related to sparse auto-encoders (SAEs)? SAEs also seem to be a popular approach for LLM interpretability. It would be good to add some discussions on this in the paper.**

**Reviewer 5PyT: ...it would be valuable to discuss Sparse Autoencoders [2] ...**

**Reviewer mxe6: The paper omits discussion of sparse autoencoders (SAEs), a prominent approach to addressing polysemanticity. It would help to situate NeuronLens relative to SAEs. In SAE-based views, disentangled concept features are represented as sparse linear combinations of neuron activations. By contrast, NeuronLens operates at the level of individual coordinates, attributing concepts to neuron-specific activation ranges. One way to reconcile these views is to note that range-based attribution can be seen as examining the per-neuron projections of a concept-direction: different concept directions induce different coordinate-wise activation distributions, which manifest as distinct ranges.**



Thank you for suggesting a discussion with SAEs. We agree with the reviewer’s viewpoint that SAE learns to represent a concept using a sparse linear combination of neuron activations, where neurons are encouraged to be of a monosemantic nature. NeuronLens can be seen as unrolling a neuron activation to identify monosemantic activation ranges, each belonging to a unique concept, with limited overlaps. NeuronLens and SAEs can be seen as complementary methods, where in the latter, SAE may serve as a first step towards disentangling the polysemanticity, and NeuronLens can further provide fine-grained monosemantic ranges. We conducted additional experiments during rebuttal to support the hypotheses and have included the discussion in the paper. We again thank the reviewer for this invaluable suggestion. We summarize the experiments below:

We conduct a preliminary experiment on concept removal in SAE features, comparing full feature blocking with activation-range masking applied to SAE features.

Since pretrained SAEs are not available for the models tested in the main experiments, we utilize pretrained GemmaScope SAEs on Gemma-2-2B. Experiments are conducted for our largest tested dataset (DB-Pedia) for two variants of SAEs.

SAEs:

1)Gemma Scope (Width_16k/l0_285)

2)Gemma Scope (Width_65k/l0_197)

Experiment Details:
We initially record the activated SAE features/neurons using samples from the training sets. We then filter these based on their frequency of activation. A feature/neuron is only considered for ablation if it activates for at least 10% for the training set for a given concept. From this subset, we select the highest activating 20% percent features and apply full feature masking or activation-range mask to test causality.  We provide the details of selected features in both settings in the table below.

| SAE        | Average Activating Features   | Unique Activating Features     | Consistent Features  | Ablated       |
|------------|--------------|---------------|---------------|---------------|
|            | (Per Sample) | (Per Concept) | (Per Concept) | (Per Concept) |
| 16k/l0_285 | 226          | 1024          | 409           | 81/16K        |
| 65k/l0_197 | 153          | 1318          | 296           | 59/65K        |
|            |              |               |               |               |

The tables below summarize the experimental results. In comparison to feature masking, activation-range masking produces a smaller reduction in complementary/auxiliary concepts and significantly lower perplexity scores, while achieving a drop comparable to full feature masking on the targeted concepts. These patterns also suggest that the SAE features are not strictly monosemantic, and they still possess neurons that capture mixtures of related concepts. While NeuronLens is effective in interpreting original polysemantic neurons, it can also be used to disentangle SAE learned representations as well.


| Method | SAE |$\Delta$ Acc | $\Delta$Conf | $\Delta$CACC | $\Delta$CCONF | $\Delta$PPL |
|---|---|---:|---:|---:|---:|---:|
| Feature Masking | 16/285 | -0.323 | -0.153 | -0.169 | -0.129 | 9.510 |
| Activation Range Masking | 16/285 | -0.322 | -0.153 | **-0.140** | **-0.095** | **0.958** |
| Feature Masking | 65/197 | -0.392 | -0.236 | -0.175 | -0.216 | 6.203 |
| Activation Range Masking | 65/197 | -0.391 | -0.235 | **-0.098** | **-0.129** | **0.335** |

---

### Decision · Action_Editor_Upcn · 2026-03-10

**Recommendation:** Accept as is

**Audience:**

Yes

**Audience Explanation:**

Yes, the problem studied in this work about neuron polysemanticity, and the interesting findings about interpretability, and LLM safety would be of interest to the general audience of TMLR.

**Claims And Evidence:**

Yes

**Claims Explanation:**

In the first round of review, one reviewer had concerns about the claims made by the authors, especially around the superiority of the NeuronLens. There were also some other concerns raised in the initial review from all three reviewers, including evidence on the core assumption, sparse autoencoders (SAEs), clarity, and evaluation issues. However, after an extensive discussion phase between the authors and reviewers, most of the concerns were well addressed, which was acknowledged by the reviewers. After the discussion, the reviewers agreed on a positive recommendation, with two Accept and one Leaning Accept ratings.

Although one reviewer still has remaining concerns about the effectiveness of NeuronLens, they agreed on the contributions of this paper, and given the additionally provided experimental evidence and the available discussions, the reviewer agrees to accept this paper.

The AE is therefore happy to recommend Accept.